# Digital mapping of surface turbulence status and aerodynamic stall on wings of a flying aircraft

Zijie Xu [1,2,5], Leo N. Y. Cao[1,2,5], Chengyu Li[1,2,5], Yingjin Luo[1,2,5], Erming Su[1,2], Weizhe Wang[3], Wei Tang [1,2], Zhaohui Yao [3] ✉ & Zhong Lin Wang [1,2,4] ✉

Real-time monitoring of flow turbulence is very difficult but extremely important in fluid dynamics, which plays an important role in flight safety and control. Turbulence can cause airflow to detach at the end of the wings, potentially resulting in the aerodynamic stall of aircraft and causing flight accidents. Here, we developed a lightweight and conformable system on the wing surface of aircraft for stall sensing. Quantitative data about airflow turbulence and the degree of boundary layer separation are provided in situ using conjunct signals provided by both triboelectric and piezoelectric effects. Thus, the system can visualize and directly measure the airflow detaching process on the airfoil, and senses the degree of airflow separation during and after a stall for large aircraft and unmanned aerial vehicles.

Quantitative analysis of airflow turbulence is vitally important for flight safety and control. Direct visualization and measurement of turbulent flow were brought into the scientific community by Reynolds' round pipe water flow experiment in 1883[1]. Navier-Stokes (N-S) equations, widely used and discussed in fluid dynamics[2,3], push turbulence research to another climax[4,5]. Especially in the field of aviation[6,7], turbulent flow has a random flow direction compared with laminar flow, leading to aircraft bumping, which affects the comfort and safety of the passengers and aircraft controllability[8]. Meanwhile, reversal turbulent flow can also separate the airflow, especially from the rear end of the wings, resulting in aircraft aerodynamic stall and flight accidents[9,10].

Thus, early warning of the airflow separation and monitoring of the stall status is critical for the safety of flying aircraft. The existing commercial monitoring systems, according to the working principles, are divided into pressure type (zeroing and non-zeroing differential pressure sensors) and mechanical type (angle of attack (AoA) indicators using wind vane). In addition to the commercial AoA indicators, there are many other emerging AoA sensors based on mechanical, electrical, optical, and thermal principles[11].

Among the commercial sensors, the mechanical AoA indicator using wind vane is most widely used in aircraft, but its drawbacks cause accidents, e.g., 346 people died in two crashes of Boeing 737-MAX 8 (2018 and 2019)[12]. In-depth analysis of its shortcomings can be summarized as follows. (i) Indirect measurement, by turning the wind vane to a fixed critical AoA position, to warn the occurrence of stall. However, the critical AoA is not a constant value, and it has a positive correlation with the flight speed and Reynolds number. Therefore, it is obvious that there are some inaccuracies in the stall warning of the wind vane method. (ii) The size and weight of the equipment. With the rise of multi-functional new unmanned aerial vehicle (UAV), it is obvious that large-size and high-quality wind vane sensors cannot be installed on these lightweight UAVs. (iii) Way to install. The installation of the wind vane sensor needs to be intrusive on the surface of the aircraft body, which can only be installed during the production of the aircraft, and later maintenance and replacement needs to be carried out by professionals. (iv) The working environment. Wind vane AoA sensing requires force to deflect wind vane so as to sense. Therefore, when the UAV flies at a low speed, it cannot work due to force problems. (v) Principles of stall sensing. The wind vane AoA sensor is

[1]CAS Center for Excellence in Nanoscience, Beijing Key Laboratory of Micro-nano Energy and Sensor, Beijing Institute of Nanoenergy and Nanosystems, Chinese Academy of Sciences, 101400 Beijing, China. [2]School of Nanoscience and Technology, University of Chinese Academy of Sciences, 100049 Beijing, China. [3]School of Engineering Science, University of Chinese Academy of Sciences, 101408 Beijing, China. [4]School of Materials Science and Engineering, Georgia Institute of Technology, Atlanta, GA 30332-0245, USA. [5]These authors contributed equally: Zijie Xu, Leo N. Y. Cao, Chengyu Li, Yingjin Luo. ✉e-mail: yaozh@ucas.edu.cn; zhong.wang@mse.gatech.edu

generally installed outside the cockpit of aircraft. It can indirectly measure the stall through AoA, which can not directly sense the airfoil air separation, and the signal generation also requires external power supply. As for the pressure-type stall sensor[13], the disadvantage is that the pressure change may or may not correlate well with the air separation and stall, making it an indirect sensing relying on back-end calculation. In addition, impurities and dirt can easily enter the system, disturbing the back-end algorithm easily[14]. Therefore, there is a great need for a sensing system with high accuracy, small size, lightweight, and no delay in visual, digital early warning of the stall by accurately delivering the degree of airflow separation during the aerodynamic stall, when the atmospheric and mechanical conditions of the airfoil are changing constantly.

Here we report a non-invasive and lightweight active system that can sense and warn the pre-stall and during stall of fixed-wing aircraft in intermediate Reynolds numbers by employing conjunct signals as provided by triboelectric and piezoelectric effects. The system is designed according to the fluid dynamics of airflow separation, making its theoretical measurement more accurate than the existing technology. We also optimize the materials (e.g., modified silk fibroin) according to the extreme flight environment around the aircraft so that it can work in cold conditions with considerable conformity and stability. Furthermore, it is waterproof and can work under high humidity and a wide range of pH conditions.

The sensing process can be divided into two stages: in the first stage, the triboelectric sensor patch generates a triboelectric signal due to the combination of the Coanda effect, airflow entrainment, and airfoil self-vibration during normal flight[15-18]; in the second stage, the sensor sheet is rolled up due to the reversal flow in the stall state. Thus, the frequency of the triboelectric signal is gradually reduced to

disappear, warning the pre-stall level, while the piezoelectric sensor patch is rolled up[19,20], indicating the severity of the stall. The triboelectric-piezoelectric conjunct sensor array with a multi-channel data receiver can be spread on the surface of the aircraft wing to construct a complete conformal, digital stall warning network. The sensor unit was tested and calibrated on the scaled-down NACA0012 model in the standard recirculation wind tunnel. Furthermore, we simulated the stall AoA curve by computational fluid dynamics (CFD) and the results help us to explore the feasibility quickly, investigate the working principle, and guide the design of the system[21-23].

In general, starting from the basis of fluid dynamics, especially the mechanism of boundary layer separation, we carefully designed the sensor with flowability by applying the customized elastic stainless steel sheet for valuable in situ airflow monitoring. System selectivity, stability, and sensitivity are also optimized from the design, structures, and materials. Most importantly, we precisely embedded the tribo-electric and piezoelectric materials in the corresponding location of the sensor to generate self-powered electrical signals, which is the core of the digital sensing and visualization system for the turbulence status and aerodynamic stall on the wings of flying aircraft.

## Results

### Working mechanism of the non-invasive digital-visualization array for turbulence stall sensing (DATSS) system

The culprit of aircraft aerodynamic stall is airflow boundary layer separation, where the flow is separated from the surface due to the reverse pressure difference of the fluid[24-26]. More specifically, the air-flow no longer adheres to the airfoil surface, causing a separation zone above the airfoil after the separation point and a large amount of reverse curling turbulence in the separation zone (Fig. 1). As a result,

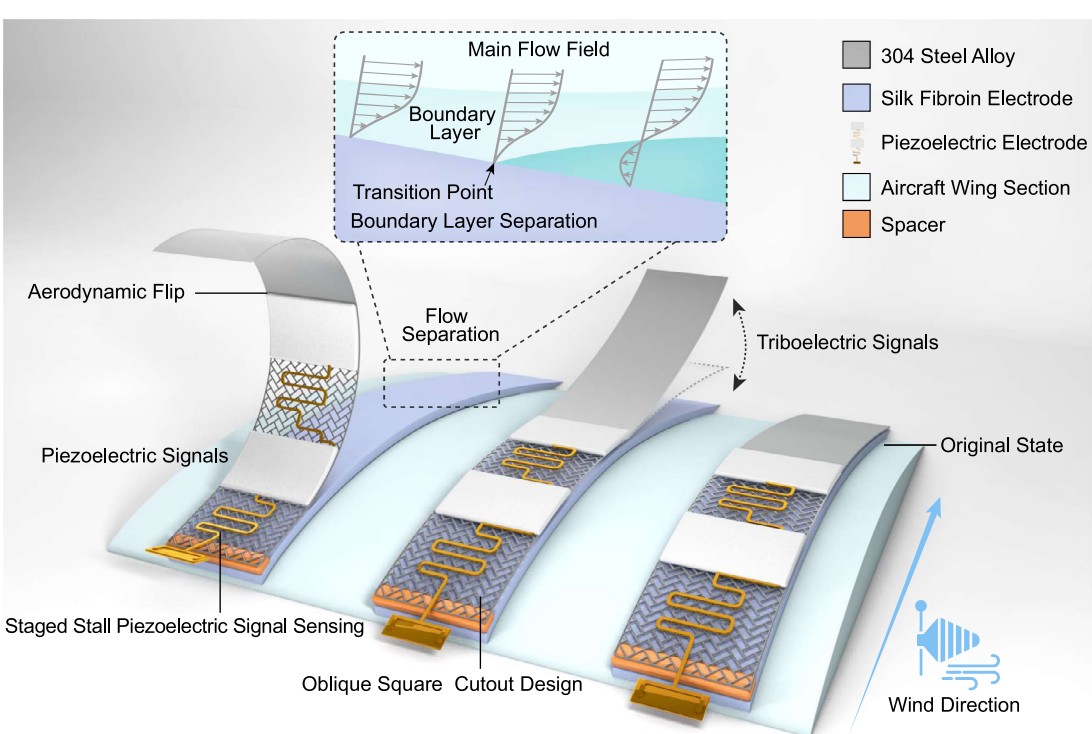

**Fig. 1 | Principle and structure diagram of digital-visualization array for turbulence stall sensing (DATSS).** The coupling of triboelectric signal (generating T-signal) and piezoelectric signal (generating P-signal) are proposed for the stall warning monitoring system. The sensing process can be divided into two stages. In the first stage, the triboelectric sensor patch generates a triboelectric signal due to the combination of the Coanda effect, airflow entrainment, and airfoil self-vibration when it is not stalled; in the second stage, the triboelectric sensor patch is rolled up due to the reverse flow in the stall state, the frequency of the triboelectric signal is gradually reduced to disappear, warning the pre-stall level, and the piezoelectric patch senses the degree of bending during the stall, indicating the severity of the stall.

the lift and lift-to-drag ratio ($K$) of the airfoil (equation (2) in Supplementary Note 1) is greatly reduced, and the aircraft enters an aerodynamic stall state. If the stall state is deep enough, a stall spin occurs, making it extremely difficult for the aircraft to recover. Today, the tuft method is still used for the airfoil airflow boundary layer separation sensing of the existing aircraft in field flight test. The tuft method can monitor the air flow separation in an array. However, its disadvantage is also obvious: without digital signals, it is impossible to store massive data, and determine the locations and the degree of air flow separation in real time.

In this work, we focus on stall sensing of aircraft, starting from the basic principles of aerodynamics. The stall sensing is achieved by directly monitoring the reversal flow and boundary layer separation, which is typical and unique during a stall, reducing false alarms compared to indirect sensors that only measure temperature and pressure. As shown in Fig. 1, the coupling of triboelectric charge separation (generating T-signal) and piezoelectric lattice polarization (generating P-signal) is proposed for the stall warning monitoring system. The system employs composite silk fibroin electrode modified by secondary structure doping with excellent robustness, coupled with triangle-arranged rectangular hollow stainless steel sheet electrode for the generation of T- and P-signals; the low-power STM-32 single-chip microcomputer system with 8 data collecting channels is used to collect, process, and transmit data.

With the above features, the main advantages of the DATSS system are as follows: (i) starting from the basics of fluid mechanics including flow reversal and boundary layer separation, the process of airflow separation can be accurately sensed in situ to give early warning of stall depth in stages; (ii) the arrayed sensing can cover the entire airfoil to directly measure (with self-powered signal generation) and visualize the stall simultaneously; (iii) highly resistant to complex environments, the system can work stably under extreme conditions such as freezing, rainy, and low temperature; (iv) the system is miniaturized and lightweight, suitable for fixed-wing large aircraft and UAVs; (v) it conforms to the surface of the aircraft and is non-invasive to the existing aircraft structure.

Here we discuss in detail how the DATSS system works. The system consists of four parts: modified silk fibroin electrode, elastic steel sheet electrode, PVDF piezoelectric electrode, and back-end data processing microcontroller. The DATSS system will collect two sets of key flight electrical signals, T-signal and P-signal. The T-signal is responsible for judging the occurrence of the stall and the P-signal is used to judge the depth of the stall.

The mechanism of T-signal generation during normal flight is described below. Due to gravity, the mounted elastic steel sheet will curve towards the curved wing surface. When the aircraft is flying at a high speed, as the airflow with a high Reynolds number passes through the steel sheet, the air will deviate from the original direction and attach to the curved surface of the sensor sheet due to the Coanda effect. The flow following the curved surface has increased velocity and, consequently, decreased pressure due to the Bernoulli principle. Thus, due to the low pressure on the upper surface, the steel sheet tends to move away from the surface of the wing. When the steel sheet and the airfoil are separated by a certain distance, the increased space between the sheet and airfoil becomes easier for high-speed airflow to enter while entraining nearby air due to viscosity, resulting in the acceleration of entrained air. According to Bernoulli's principle again, the region of accelerated air will have lower pressure compared with the low-velocity air (atmospheric pressure) along the same streamline, so the low-pressure area created between the steel sheet and the airfoil will give the steel sheet a tendency to move close to the airfoil. Together with the help of steel sheet elasticity and natural vibration, the above process of separation and reattachment can repeat continuously and the steel sheet will tap rhythmically on the airfoil, generating a constant T-signal during normal flight.

The mechanism of P-signal generation during aircraft stall is described below. As the flow passes the curved airfoil, over the front portion of the airfoil, the pressure decreases along the flow—a favorable pressure gradient; over the rear portion of the airfoil, the pressure increases along the flow—an adverse pressure gradient. As the AoA increases, the adverse pressure gradient increases. Meanwhile, the normal fluid flows against the pressure, until it can't and separates from the surface, increasing the pressure drag of the aircraft drastically and leading dynamic stall. Meanwhile, in the zone near the surface, the large adverse pressure gradient reverses the flow, intending to flip and push the steel sheet against the flow direction. Taking advantage of the intention, we applied the P-signal materials to the bending location of the sensor sheet, where P-signal can be generated due to pressing and bending from the reversal flow. We discuss the DATSS system aerodynamics supplement description in Supplementary Note 2 and mechanics principle of T/P-signal generation of DATSS system in Supplementary Note 3.

In addition, the signal generation and sensor structure are optimized by hollowing the elastic steel sheet with triangle-arranged rectangular patterns, inspired by the ancient window structure in the East. This hollowed structure can balance the pressure difference on both sides of the sheet for normal flight conditions so that the sheet will not flip due to the light adverse pressure gradient, reducing the chances of false alarms. Meanwhile, the triangular arrangement maintains the mechanical strength of the structure, while making the sensor more sensitive to the flip force and making P-signal more responsive when the stall happens. At the same time, T/P signals are analyzed and discussed in depth in Supplementary Note 5. With the careful investigation of sensing selectivity, stability, and sensitivity, this study proposes a new turbulence stall warning system for the study, design, and safety monitoring of future aircraft.

## Materials optimization of DATSS

T-signal and P-signal are generated using modified silk fibroin and elastic steel sheet electrodes, and PVDF piezoelectric electrodes, respectively. Since the triboelectric effect is ubiquitous, T-signal has a wide range of material selection, resulting a lot of room for material optimization. The candidate material needs to maintain its performance and stability in the complex flight environment (e.g., low-temperature icing conditions and high humidity), and be conformal to the airfoil. Modified silk fibroin has been explored recently and showed its possibility as a good candidate for the sensor electrode in complex environments. In fact, modified silk fibroin has been already applied to aerospace suits[27]. In addition, the silk fibroin material is easy to lose electrons in the triboelectric signal sequence to form a stable T-signal system with an elastic steel sheet.

A critical process for the silk fibroin to own excellent mechanical and physicochemical properties is mesoscopic manipulation since unmodified natural silk fibroin is with relatively poor mechanical properties and thermal stability, and low corrosion resistance[28]. Through modification and doping, the secondary structure of silk fibroin can be transformed. Thus, the electrode made from the modified silk fibroin has excellent conformity and transparency, stable electrical signal, easy separation and analysis, low-temperature resistance, acid and alkali resistance, stretchability, and excellent water resistance, making it a good candidate material for aircraft stall sensing.

The detailed mechanism and process of the modification are as follows. Natural silk fibroin is bound by silk fibroin through weak intermolecular interactions, including four different secondary structures: α-helix, random floc, β-sheet, and non-covalent binding. Since its mechanical flexibility and chemical stability mainly depend on the intermolecular interactions of silk fibroin, it is of great significance to control the mesoscopic structure and enhance the intermolecular force by doping. In this work, we use polyhydroxy functional group

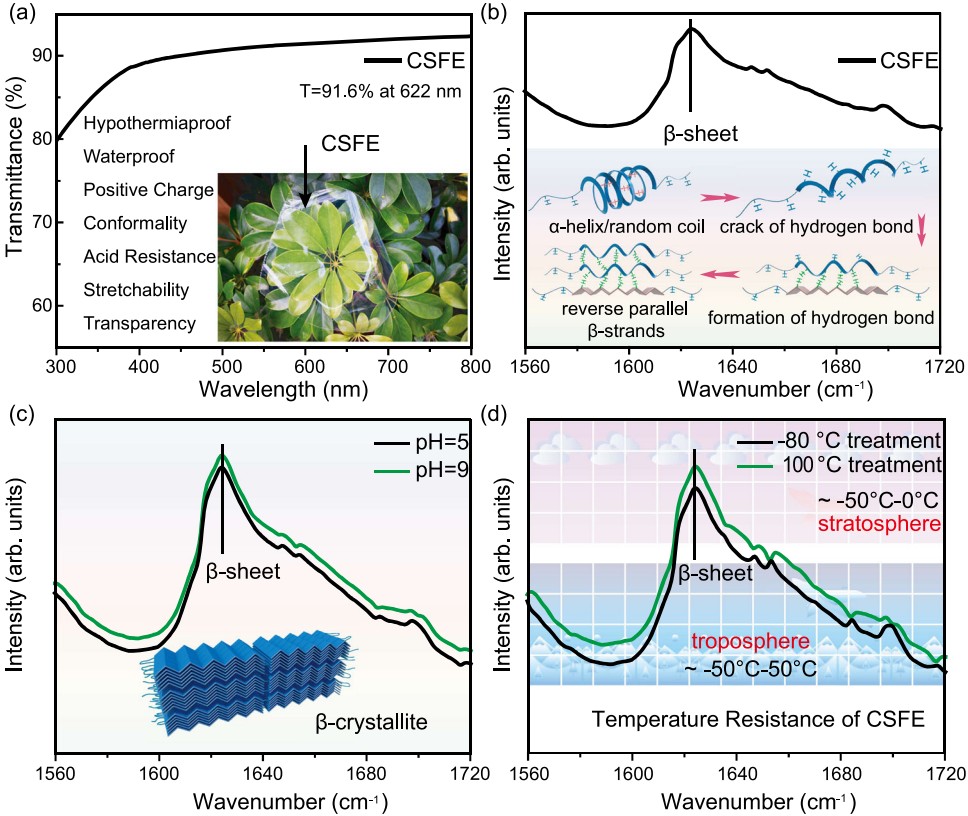

**Fig. 2 | CSFE material performance characterization test. a** Light transmission performance test of CSFE material, the inset is a photograph of CSFE with $T$ = 91.6% at 622 nm. **b** Infrared spectrum of CSFE material, the inset is a schematic diagram of the structure for the transition of the secondary structure of silk fibroin. **c** Infrared spectra of CSFE in pH 5 and 9 environments. **d** Infrared spectra of CSFE after treatment at −80 °C and 100 °C.

dopants for secondary structure modification (Supplementary Fig. 1), whose detailed doping process and mechanism are discussed in the Methods section, since each hydroxyl structure tends to break its α-helix structure by breaking hydrogen bonds after adding silk fibroin to form a more stable β-sheet structure[29].

The light transmittance of the modified composite silk fibroin electrode (CSFE) is greater than 80% in the range of 300 nm-800 nm and is 91.6% at 622 nm (Fig. 2a). The photograph of the modified silk fibroin electrode in this work is shown in the inset of Fig. 2a. Six advantages of CSFE in DATSS system are discussed as follows:

(i) Transparency: considerable light transmission performance can be adapted to the aircraft wing surface without changing the original appearance of the aircraft. (ii) Conformality: due to the aerodynamic design, the electronic sensor with excellent conformality can adapt to various aircraft wing surfaces. (iii) Mechanical strength: robustness is a key issue in engineering design, especially in complex flight environments. The modified CSFE exhibited the β-sheet characteristic peak of the Fourier transform infrared (FTIR) absorption spectrum (Fig. 2b), meaning the structure is transferred from the α-helix to the β-sheet position which is more stable. (iv) Acid and alkali resistance: since the stability of steel sheet alloy material in the weak acid and weak alkali environment has been confirmed, we mainly consider the acid and alkali resistance of modified CSFE. Here we tested the acid and alkali resistance of the electrode at pH values of 5 and 9. After 24 h of immersion treatment, no obvious change in its structure was found in the FTIR absorption spectrum (Fig. 2c). (v) Temperature resistance: for aircraft operating in the troposphere and stratosphere, the ambient operating temperature is between −50 degrees Celsius (−50 °C) and 50 °C. However, considering the extreme cases, we tested the FTIR absorption spectrum of CSFE in the environment of −80 °C and 100 °C, and the characteristic peaks of its

structure did not change significantly (Fig. 2d). (vi) Stretchability: The tested tensile strength is greater than 150% (Supplementary Fig. 2). This result is superior to that of the most commonly used polymer materials, helping it better fit the aircraft wing surface. vii) Signal strength: for the research of T-signal, Wang's group has carried out a comprehensive exploration of its theory and engineering application since 2012 so that this work has a technical basis for the stability and collection of T-signal[30].

As a positive triboelectric material, modified CSFE is excellent as a positive triboelectric material with superior comprehensive advantages over most polymer materials. Interestingly, we also tested the performance of negatively charged polymer triboelectric materials for signal generation. Under the high airspeed and the corresponding high flapping frequency, negatively charged polymer triboelectric materials easily attracted the steel sheet due to the final quantity of accumulated negative charges, inhibiting its flapping effect. In contrast, the modified CSFE avoids this adhesion problem. At the same time, the influence of DATSS system on lift and drag of aircraft airfoil is discussed in Supplementary Note 4.

**Validation and calibration of DATSS**

The test in the standard recirculation wind tunnel is extremely important to calibrate and evaluate the performance of the DATSS system. In this work, all wind tunnel data are collected from the recirculation wind tunnel (Prandtl−500). The side perspective view of the wind tunnel schematics is shown in Fig. 3g. Its total length is 14.37 m, the height is 4.81 m, and the total length of the test section is 2.02 m. The relationship between wind speed and wind tunnel motor speed is shown in Supplementary Fig. 3. The parameters of NACA0012 airfoil used in this work are shown in Supplementary Table 1. T/P-signals in this section are collected separately using an electrometer.

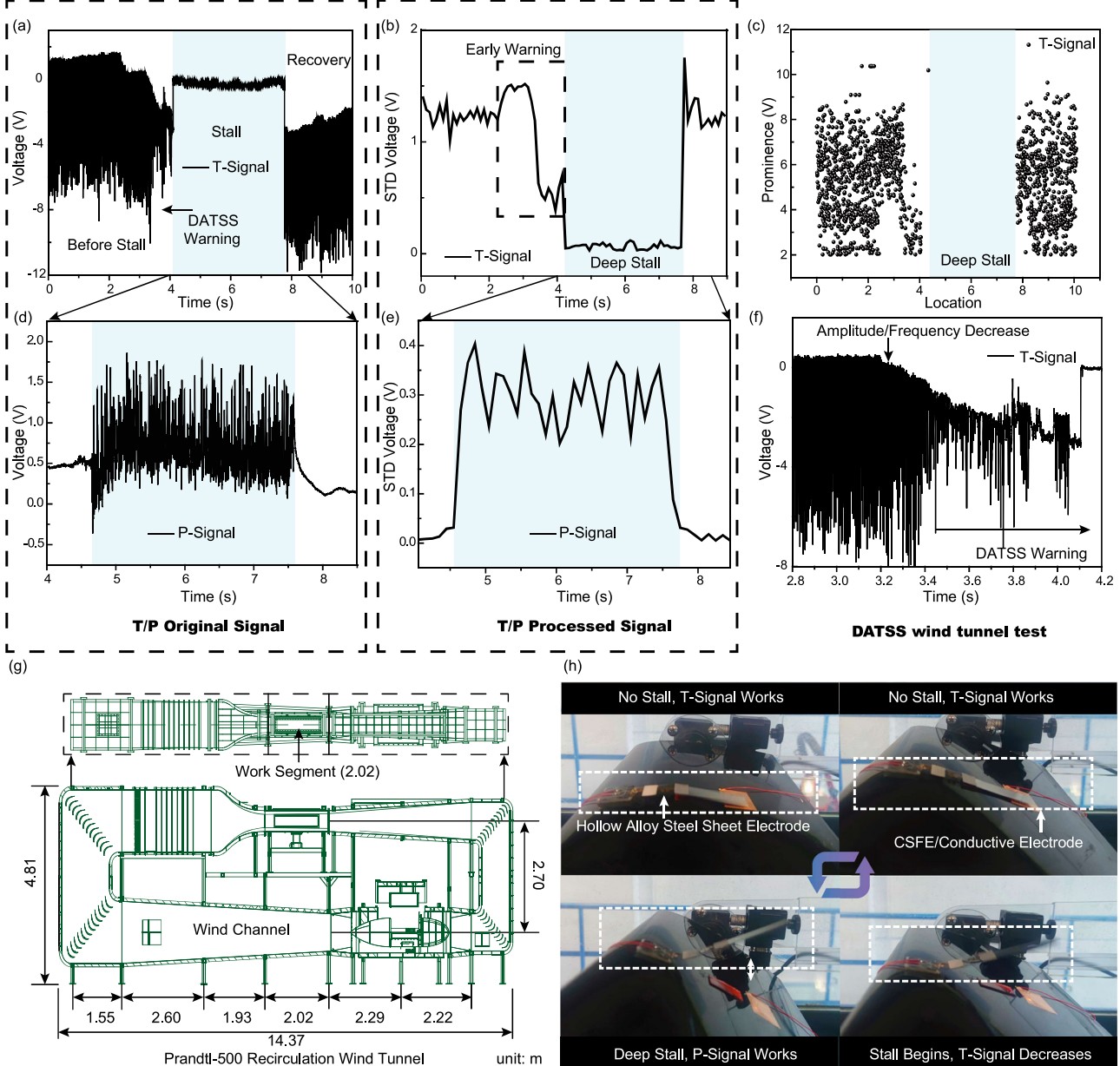

**Fig. 3 | Performance test of active DATSS in recirculation wind tunnel. a** DATSS system stall sensing raw T-signal. **b** DATSS system stall sensing T-signal after STD processing. **c** DATSS system stall sensing T-signal after prominence (function of findpeaks in MatLab) processing. **d** Variation of T-signal at imminent stall. **e** DATSS system stall sensing raw P-signal. **f** DATSS system stall sensing P-signal after STD processing. **g** Schematic diagram of the structure of the Prandtl−500 recirculation wind tunnel, which is divided into two floors, the work area is located on the second floor. **h** Photograph of DATSS system visualizing the entire process of sensing the stall.

Here we discuss the measurement in three stages before stall, during stall, and after recovery (Fig. 3a). The test wind speed is selected as 40 m s$^{-1}$, which is the cruising speed of a small aircraft and UAVs. When the aircraft does not stall, the T-signal fluctuates obviously. When the aircraft is close to stall due to the increase of AoA, the airflow is separated and the influence of the airflow separation effect is much greater on the sheet than the flutter effect, reducing the amplitude and frequency of the T-signal significantly (Fig. 3f). When the aircraft AoA continues to increase to a complete stall (deep stall), the T-signal almost disappears, which is significantly different from that before the stall, so that the pre-stall is clearly sensed by the T-signal. During the stall, due to the curling of the reverse airflow, the system is then dominated by the P-signal. We use two-phased P-signals to reflect the extent of flow reversal and stall. After that, as AoA decreases until the

stall is successfully recovered, the airflow separation disappears and the T-signal is restored. We also tested the T-signal test for the wind speed of 60 m s$^{-1}$ (Supplementary Fig. 4).

For the data analysis, the DATSS system will not rely on complex algorithms as existing methods so that our data processing is logically simpler. Instead, we used a simple standard deviation (STD) method to process T/P-signals. For the T-signal (Fig. 3b, c), the initial data in Fig. 3a were simply calculated with standard deviation (STD) for every 1000 data points, generating the secondary data in Fig. 3b. Since the STD method only records relative change between adjacent data points, the signal baseline drifting (problematic in many instruments and measurements) is minimized. The colored region portion in Fig. 3b is a significant stall region. Right before the occurrence of the stall, the results showed a sharp drop to the vicinity of zero value, through

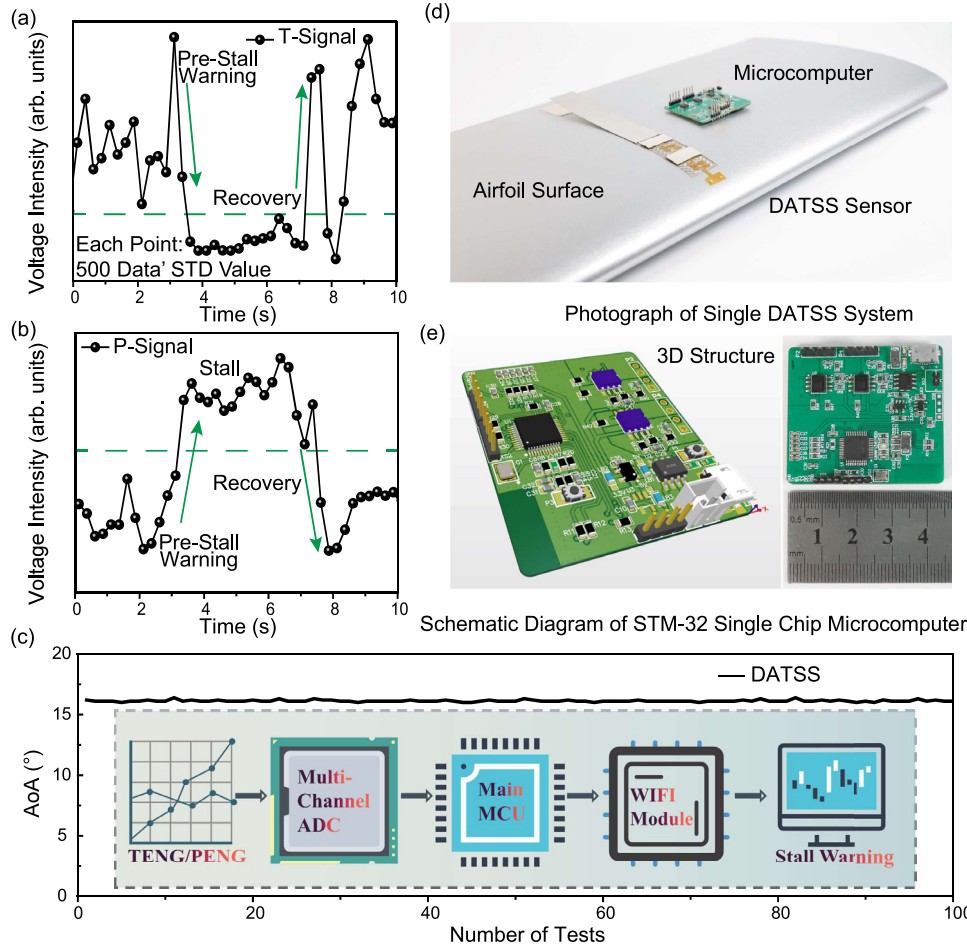

**Fig. 4 | Development and testing of DATSS wireless signal sensing system. a** The T-signal wirelessly transmitted by the DATSS system (processed by STD algorithm: one STD data point from every 100 real-time data points), which can intuitively and accurately show the stalling process. **b** The P-signal wirelessly transmitted by the DATSS system (processed by STD), due to the influence of the complex high-speed turbulent wind field, the P-signal has hysteresis compared with the T-signal. **c** The relationship between 100 times stall and AoA, the inset shows the working principle of the wireless signal transmission. **d** The photograph of the DATSS system customized NACA0012 airfoil in the wind tunnel. **e** The schematic diagram of the STM-32 microcontroller, its side length is about 4.5 cm.

which we can clearly warn the stall. We use the similar method to handle the P-signal (Fig. 3d, e), which has presented the same significant stall warning. Before the stall, the P-signal only has a small fluctuation with background noise; when the stall occurs, the amplitude of the P-signal suddenly increases. Note that the results for the stall and non-stall are completely different so that the accurate stall sensing does not depend on how quickly the stall happens (which may not be the case for pressure-based skin sensors).

Another data processing method is also proposed. As shown in Fig. 3c, the method of prominence (function of findpeaks in MatLab) measures how much a peak outstands due to its intrinsic height and location relative to other adjacent peaks. A threshold of 0.5 for minimum prominence is applied in our study so that noise prominence smaller than 0.5 will be excluded. We can clearly observe from Fig. 3c that in the black-dot concentrated zone, the aircraft does not stall and the yellow area with almost no black dots is a stall zone. This method can directly pass the data map to the pilot or UAV operation personnel in real-time to warn the whole process of the stall, with minimum data reprocessing. P-signal after prominence processing is also exhibited (Supplementary Fig. 5). Besides, the frequency of prominence is also shown in Supplementary Figs. 6, 7.

In Fig. 3h, we demonstrate the visualization of DATSS system in the turbulence stall. In the wind tunnel, when the aircraft AoA is small, the airflow on the airfoil surface does not separate and the T-signal continues to work; as the AoA increases, the T-signal gradually

disappears, the P-signal begins to increase, and the sensor sheet rolls over, revealing the occurrence of stall visually.

## Development of wireless modular for DATSS system

In this work, we have developed an automatic wireless modular using STM-32 low-power microcontrollers to receive, process, and transmit data from DATSS in real-time. This will allow the DATSS system to better enter industrial mass production. The photograph of the DATSS system is shown in Fig. 4d. Figure 4a, b exhibits the T-signal and P-signal collected and processed by the single chip microcomputer, respectively. The single-chip microcomputer can also transmit the results to the terminal via WiFi. In Fig. 4c, we use the DATSS system to accurately predict the occurrence of 100 times stalls under the wind speed of 40 m s⁻¹. Figure 4e exhibits the schematic diagram of the STM-32 microcontroller. Its side length is as small as about 4.5 cm and the total mass is as light as 8.2 g with eight test channels, which can meet the array sensing requirements of the DATSS system. The detailed design scheme is described in the Methods section. Furthermore, we discuss the accuracy the DATSS system in depth in Supplementary Note 7.

## Field test of DATSS wireless system and CFD simulation

After completing the standard wind tunnel test of the DATSS system, we mounted the DATSS system on a small fixed-wing aircraft for real-world field tests. We used a fixed-wing remotely controlled Cessna 182

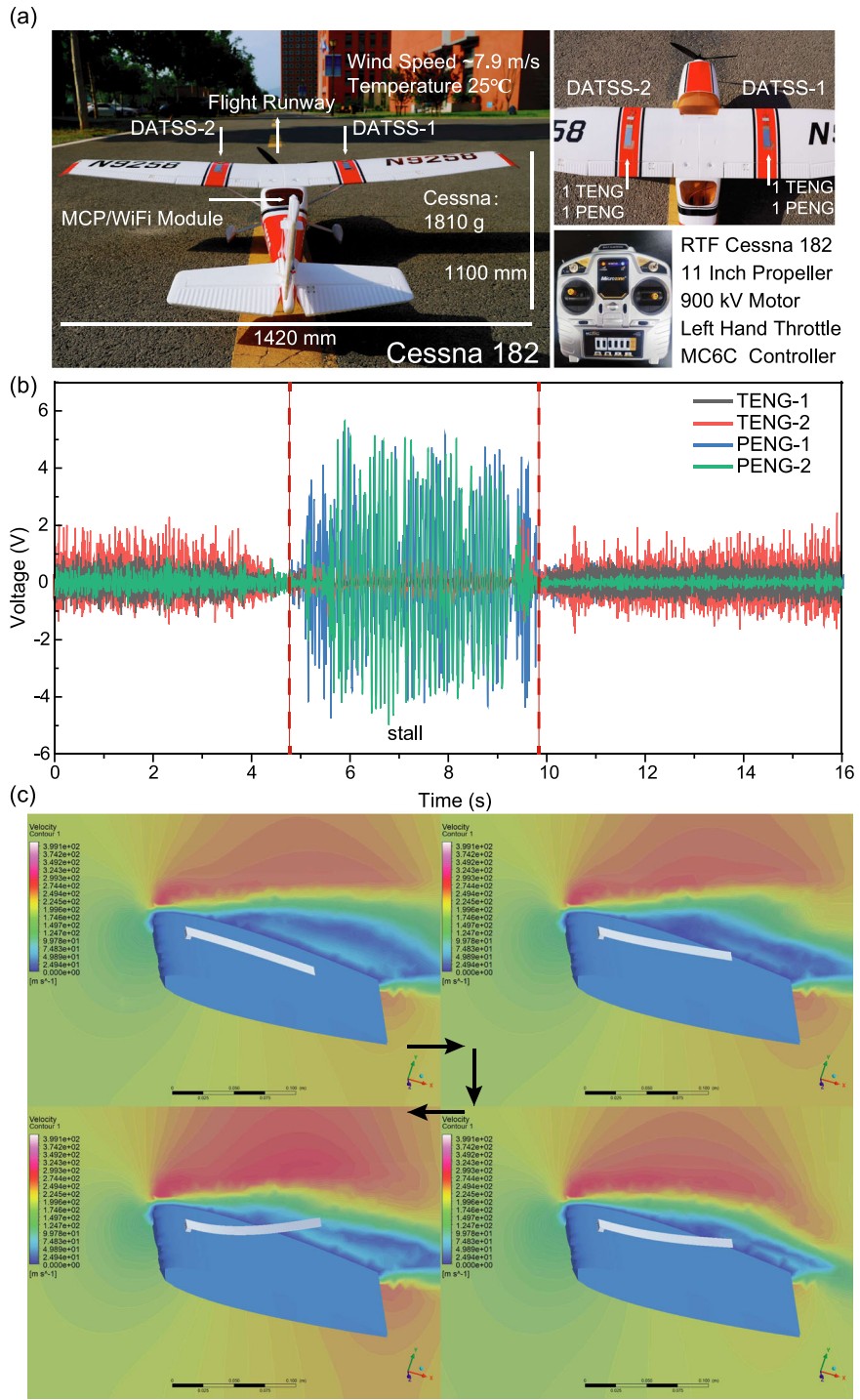

**Fig. 5 | Flight test of DATSS system and CFD simulation. a** The photograph of fixed-wing remote control Cessna 182 aircraft with a wing length of 1420 mm for flight test, equipped with two DATSS units. **b** The wireless flight signal of the arrayed DATSS system. **c** CFD simulation of fluid-structure interaction of DATSS system when the AoA of the aircraft is 18°.

model aircraft (with a wing length of 1420 mm), equipped with two DATSS units on each wing surface (Fig. 5a). Each DATSS system includes one T-signal and one P-signal generator. Figure 5b shows the wireless signal of the arrayed DATSS system. We find that due to different testing environments, the T/P-signals have different amplitudes compared with wind tunnel data. However, the stall warning and stall depth analysis still shows the same trends: right before the stall occurs, the two T-signals start to weaken and the P-signal is enhanced. Note that the signal amplitude for DATSS-2 is higher during the stall process

than DATSS-1, namely, the realistic stall depth on the wings of the aircraft could be different depending on the location on the wing and environmental factors, further illustrating the point of the arrayed system. The successful collection of data from the field experiment of the arrayed DATSS wireless system can provide a digital and visualized stall sensing solution for the design and flight stages of various fixed-wing aircraft in the near future. At the same time, large Reynolds number experiments and limitations of DATSS systems are discussed in Supplementary Note 6.

During the development of the DATSS system, we conduct a large number of CFD simulations to verify its theoretical feasibility and optimize the system performance. Figure 5c completely shows the whole process of 3D two-way fluid-structure interaction CFD simulation of DATSS system. A continuous T-signal is generated at AoA of 0°, a continuous P-signal is generated at 18°, and the T-signal is lost. This result is consistent with the conclusion of wind tunnel experiment and can be used as an important basis for theoretical explanation of the working principle of DATSS system. The detailed simulation process and discussion are described in the Methods and Supplementary Note 9. In order to explore the mechanical principle of DATSS, we use CFD to simulate the flow field around the airfoil surface as the AoA of the aircraft increased from 0° to 24° (Supplementary Note 8/Supplementary Video 1–5). We use arrows and colors to mark the velocity direction and magnitude of the flow field, respectively. When the AoA gradually increases, the separation of the surface airflow can be observed after about 12° and part of the airflow (direction indicated by arrows) begins to move in the opposite direction of the incoming flow. When the AoA continues to increase, the separated airflow begins to roll and reciprocate (after about 16°), enhancing the P-signal. In general, CFD simulations help better explain the working mechanics and guide the design of the DATSS system.

In order to verify the applicability of the DATSS system at large Reynolds numbers and the feasibility of the DATSS system during the flight of a real manned vehicle, the sensing test of a real Cessna C172S aircraft was carried out. It is worth noting that the DATSS system of manned spacecraft was tested with the same trend as that in the wind tunnel. The detailed flight test process and result analysis are described in detail in Supplementary Note 10 (DATSS system for Cessna C172S manned vehicle testing). In general, the DATSS system was designed from the fundamental of aerodynamics, optimized on the conjunct triboelectric/piezoelectric signal, evaluated in the intermediate Reynolds number wind tunnel, and field tested with the manned aircraft, making it a promising strategy for the new in situ stall sensing.

## Discussion

In summary, we proposed a digital-visualization array for turbulence stall sensing (DATSS) system, combining active triboelectric and piezoelectric effects, to visually and digitally map the in situ surface turbulence and aerodynamic stall of flying aircraft. The sensing is achieved by monitoring the motion of the floating sensor sheet driven by the turbulent flow right before and during the aerodynamic stall: the sheet flapping before the stall generated triboelectric signals; the sheet flipping due to the reversal flow generated graded piezoelectric signals.

The sheet design started from the fundamental of aerodynamics, taking advantage of the Coanda effect, Bernoulli principle, air entrainment, and boundary layer separation. The designed sensing system was then rigorously calibrated and validated by the standard recirculation wind tunnel and airfoil model testing, and field measurement, assisted by dynamic simulation using computational fluid dynamics (CFD). The system is also designed with highly industrialized manufacturing, allowing the DATSS system to well enter the industrial production stage.

Targeting broad practical applications, the system is optimized in terms of (1) structures: miniaturization, non-invasive, minimum interference with the main flow, and sensitive to the interested airflow, and (2) materials: various resistances including temperature, acid, alkali, and water. Since the selection of materials for the triboelectric effect that is ubiquitous is super wide, the adaptability of our sensor to the specific surrounding environments can be further improved. Furthermore, by adding a low-power microcomputer modular, the system can collect, process, and wirelessly transmit data in real-time to the pilot for early warning and precaution, taking the practicality one step

further. In general, this study has explored the feasibility of triboelectric and piezoelectric effects on fluid dynamics sensing for flying aircraft and ushered in a new era in digital turbulence mapping.

## Methods

### Preparation and secondary structure modification of CSFE

Silk fibroin materials have been already used in aviation and aerospace, especially in the field of wearable fabrics. These silk fibroin materials are natural silk fibroin after modification, whose mechanism varies depending on the composition of dopants or modifiers in various works. In the past nearly ten years, we have explored a large number of studies related to silk fibroin modification[28,29,31–34]. Among them, the research on the improvement of its mechanical and chemical properties enables silk fibroin materials to be applied in a wider field. In this work, 15 wt% waterborne polyurethane (WPU) was selected as the dopant. Furthermore we provided a theoretical explanation for its secondary structure modification mechanism.

At first, CSFE was prepared by the following steps: added $Na_2CO_3$ (0.05 M) into boiling distilled water to dissolve completely, then added the pre-cleaned cocoons to the boiling solution, and washed for 30 min to remove the useless structure of the silk (degumming treatment). This process was repeated four times. Put the treated silk into distilled water at 60 °C for washing, repeated 3 times to remove $Na_2CO_3$, and then placed the degummed silk in an oven at 60 °C to dry. Prepared lithium bromide solution (9.3 M), then fully immersed the degummed silk in the lithium bromide solution, and placed it in an oven at 60 °C for 6 h until the silk fibers were completely dissolved to obtain the silk fibroin solution. The silk fibroin solution was put into a dialysis bag (molecular weight 3500) and dialyzed in deionized water at 25 °C for 72 h. The water was changed every 4 h and then centrifuged at 8000 r min$^{-1}$ for 20 min. 15 wt% WPU and 5 wt% glycerol were mixed with the centrifuged silk fibroin solution (supernatant) and dried at room temperature to form CSFE. According to our previous work, dopants rich in hydroxyl functional groups will significantly enhance the mechanical and chemical properties of CSFE. 15 wt% WPU was introduced to enhance the performance of CSFE by mesoscopic doping of silk fibroin solution. Considering the complex working environment of the aircraft, we balanced the relationship between the electrode size and durability by increasing the doping amount of WPU paper by 15 wt%.

Here we provide a theoretical explanation for the mechanism of secondary structure transformation in silk protein when adding WPU into pure SF as shown in Supplementary Fig. 1. The hydroxyl group in WPU obviously played a role while the rest of the functional groups in WPU did not to promote the conversion of α-helix to β-sheet.

The nucleation kinetics of α-helix to β-sheet was also reported. The urethane WPU grid chain structure contains a large number of hydroxyl groups. After adding it to silk fibroin, the hydroxyl groups tend to break the original bond of the α-helix structure. After that, the broken α-helix unit will be in a free state, which is inclined to form a new hydrogen bond with a new receptor, and at the same time, the mesh chain structure of WPU can provide a new stable position for broken α-helix unit, forming a new hydrogen bond to transform it from a free state to a temporary stable structure on a fixed orientation. After this process, a new hydrogen bond is formed and the fixed α-helix unit tends to absorb another free α-helix unit, re-nucleate to form a reverse β-strand, and then convert to a more stable β-sheet structure. After this, a number of transformed stable β-sheets form β-crystals by stacking. Since β-sheet can be more stable in the air or water environment, the mesoscopic doping of regenerated silk fibroin has significantly improved stability.

In addition, glycerol (Gl) is introduced to strengthen the waterproof performance of the film without changing the structure as previously reported. The prepared modified silk fibroin was subjected to magnetron sputtering with an ultra-thin silver conductive layer

(18 μm). The aforementioned experimental results show that, CSFE owns low temperature resistance, acid and alkali resistance, stretchability, excellent water resistance, excellent light transmission and conformal properties, Besides, after losing electrons, it forms a stable and durable T-signal sensing electrode with the sensor steel sheet.

### DATSS system integration

The DATSS system consisted of four parts: CSFE, triangle-arranged rectangular hollow alloy steel sheet, PVDF piezoelectric electrode, and data processing microcontroller. The preparation process and characterization of CSFE have been discussed in detail in the last section. In this section, we focused on the preparation of alloy steel sheets and PVDF piezoelectric electrodes. Considering the complex environment in which the aircraft flies, the only moving electrode in the DATSS system requires durability and stability. 304 stainless steel, a widely used chromium-nickel stainless steel, had considerable corrosion resistance, heat resistance, low-temperature strength, and mechanical properties. Ultra-thin 304 alloy steel sheet (0.1 mm thickness, 2.2 cm width and 10.0 cm length) was the ideal counter electrode choice for T-signal of this work. According to Eq. (1) of the stiffness of the steel sheet in theoretical mechanics, the stiffness ($M$) of the steel sheet is positively correlated with the width ($b$) of the steel sheet, the cube of the thickness ($h$), and its elastic modulus ($E$), and negatively correlated with the working length ($L$) of the steel sheet. Therefore, when the thickness of the steel sheet was doubled, its stiffness will increase eightfold. While ensuring the strength of the T-signal, reducing the thickness of the steel sheet can effectively reduce its stiffness and provide a dynamic basis for its visualization of flipping; simultaneously, it also made the DATSS system more lightweight, reduced the mechanical damage to CSFE electrodes, and increased the durability of the system.

$$M = \frac{Ebh^3}{12L} \tag{1}$$

In addition, triangle-arranged rectangular hollow structure was applied to the length of the first half of the alloy steel sheet (the length of the DATSS system can vary based on different airfoils). The details of PCB design for graded P-signals and the hollow structure were shown in Supplementary Figs. 8 and 9. The alternating arrangement of the triangles enhanced the mechanical strength of the steel sheet during the flip process. The rectangular hollow pattern allows the sensor sheet response easily to the reverse separation airflow, while reducing the interference of the DATSS system to the airflow field. Since the bending of the steel sheet occurred on its front side during the flipping process, we chose to hollow out the first half (close to the pivot), and stably generated the T-signal in the second half. By using two PVDF electrodes (for graded signals) and a gold-plated PCB circuit board as the generation and collection device of the P-signal, the graded signal can effectively judge the depth of the stall. The total length of the PCB circuit board was 59.7 mm. The S-shaped circuit can play a role in anti-stretching during the rolling process. The PVDF electrode area was 20.3 mm long and 10.0 mm wide. The leading edge between the steel sheet and the airfoil was separated and fixed with 3 M foam spacer (2.2 cm wide, 0.5 cm long and 0.3 cm height), so that the Coanda effect, entrainment, and vibration are effectively generated[35]. The entire DATSS system architecture was shown in Fig. 1. Besides, the comparison of T-signal of hollow alloy steel sheet material coupled with different materials was displayed (Supplementary Fig. 10).

### DATSS system wind tunnel testing

In this work, all stall-related data were tested using a recirculation wind tunnel (Prandtl-500). The total length of the wind tunnel was 14.4 m and the height was 4.8 m. The wind tunnel had a 1.9 m long neck section to reduce the boundary layer effect of the working section. A 2.6 m

filter section was configured in front of the constriction section of the wind tunnel, which can play the role of filtering the flow field. The total length of the working section was 2.0 m, equipped with a three-sided glass observation area, and the bottom can be connected to a variety of data testing equipment in this work. The maximum wind speed can reach 80 m s⁻¹. Supplementary Table 2 exhibited the calculation of Reynolds number at different speeds in the wind tunnel test. In the wind tunnel, the turbulence of the incoming flow is less than 0.2%, and the change rate of the uniformity of the wind speed at each point within a section of 500 mm×500 mm is less than 0.5%. On the upper surface of the working area of the wind tunnel, there was an opening area that can simulate different pH values of rainwater, different humidity and other test environments. In addition, we customized the aircraft wing segment with adjustable AoA via stepping motor control according to the experimental needs. The AoA adjustment accuracy was 0.1°, and the maximum adjustment speed was 10° s⁻¹. The stepping motor was controlled by an external computer, and the T/P-signal data was obtained by two test methods: Keithley 6514 electrometer and low-power STM-32 microcontroller. The entire wind tunnel is designed with two floors, and the experimenters use high-speed cameras on the second floor to take experimental photos in this work. Stall sensing T-signal of DATSS system under surface icing and rain condition were shown in Supplementary Figs. 11 and 12. The whole process of DATSS system working in wind tunnel was shown in Supplementary Video 6.

### 3D CFD simulation

Supplementary Video 7 shows the whole working process of DATSS system when AoA=0° and 18°. Ansys Fluent was used to simulate the working process of DATSS system. NACA0012 three-dimensional airfoil was employed to simulate. The simulation used two-way fluid-structure interaction simulation, detailed settings and calculations were in the Supplementary Note 9.

### Field experiment design and data collection

Development and design scheme of wireless signal transmission system: the wireless signal transmission system in this work adopted multi-channel wireless signal acquisition circuit system, which consisted of charge preamplifier, 8-channel A/D conversion chip, main control STM-32 chip, WiFi and power management module. The charge preamplifier collected, regulated, and amplified the T/P-signal, and then sent it to the A/D conversion chip to digitize the analog voltage signal. Then, through the control of the STM-32 chip, the data was transmitted to the WiFi module through SPI communication, and to the mobile terminal through wireless WiFi to realize real-time sensing and monitoring. The multi-channel wireless signal acquisition circuit was developed and designed by Altium Designer, and the program code was developed and written by MDK-ARM.

The flight experiment of the DATSS system was carried out in the closed field of the Chinese Academy of Sciences. The experimental wind speed was about 7.9 m s⁻¹, and the experimental environment temperature was 25 °C. The aircraft was a Cessna 182 model aircraft with a wing length of 1420 mm and a fuselage length of 1100 mm. The total weight of the aircraft was 1810 g, equipped with 900 kV motor, MC6C left-hand throttle controller. The aircraft was divided into 5 communication channels, distributed control of two main wing flaps, tail elevator/horizontal rudder and throttle. Two DATSSs (using the DATSS-1/2 designation) were installed on each side of the main wing, each DATSS system consisted of a T-signal sensor and a P-signal sensor (due to the low wind speed, no hierarchical P-signal sensor was required).

### Characterizations

All airfoil segment stall data in this work were measured using the Prandtl-500 recirculation wind tunnel (Tianjin Prandtl Co., Ltd.). The T/P-signal was measured by Keithley 6514 electrometer, and the

wireless T/P-signal was collected by STM-32 low-power micro-controller (self-developed programming). The parameters of the air-foil segment were obtained by machining stainless steel with CNC lathe, the chord length was 20 cm, and the AoA was controlled by a stepper motor (Cangzhou Kederuier Experimental Equipment Co., Ltd.). The piezoelectric electrode used commercial silver-coated PVDF electrode (Zhimeikang Technology (Shenzhen) Co., Ltd.). The spacer employed 3 M double-sided adhesive tape (total 3 mm high). The 304 hollow steel sheet was prepared using a commercial laser cutting machine (Yongbin Metal Technology), which also cut and shape the polymer material templates required for this work. The PCB circuit board was designed with Altium Designer software, and prepared on the PI flexible substrate using copper deposition technology. We pre-pared CSFE electrodes using commercial natural silkworm chrysalis material (Beijing Lablead trading Co., Ltd., China) and other reactants were purchased in Aladdin group. The infrared spectra of CSFE were measured by Fourier transform microinfrared spectrometer (Nicolet IN10). The photographs we displayed were taken by a Nikon camera (D5300). Tensile testing of CSFE was tested by a tensile tester (micro tester 5948). Magnetron sputtering was employed to plate electro-conductive material on the surface of CSFE under a three-target sputtering device (MSP-620 automatic sputtering) and the target material was purchased from Zhongnuo Advanced Material (Beijing, China). The transmittance of the CSFE was measured by UV–VIS spectroscopy instrument (Lambda 750). Deionized water was obtained by an ultrapure water machine (Jingyi, China). Petri dishes (9 cm in diameter) and centrifuge were purchased from Labelad (Beijing Labead trading Co., Ltd., China). The reactants were weighed using Yingheng Technology YHX1 electronic balance. The pH measurement was employed Xima PH828+ pH meter. 1.4 m fixed-wing electric model aircraft was purchased from Yigou Aviation Technology.

## Data availability

Source data are provided with this paper.

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

## Acknowledgements

The research was supported by the National Key R&D Project from the Minister of Science and Technology (2021YFA1201601); National Natural Science Foundation of China (Grant Nos. 51432005, 51561145021, and 52203324); the Fundamental Research Funds for the Central Universities (E2E46804). The authors also thank the technical support from Prof. Hongwei Wang, Zhuyu Zhou, Hengrui Sheng, Jianzhe Luo, Dr. Yujia Lv, and Wen Dong.

## Author contributions

Z.L.W. and Z.Y. supervised the project. Z.L.W., W.T. Z.X., and L.N.Y.C. conceived the idea, designed the experiments, and wrote the manuscript. Z.X. synthesized and modified silk fibroin electrodes, and characterized them. L.N.Y.C., E.S., Y.L., and W.W. carried out fluid mechanics and wind tunnel experiments, and CFD and theoretical calculations. C.L. designed wireless modular, conduct modular data collection, and optimizes the DATSS system. Z.X. and L.N.Y.C. collected and optimized various data for DATSS system, including the field test. Z.Y. and W.W. provided technical support and theoretical guidance for wind tunnel experiments. L.N.Y.C. and Z.X. discussed the aerodynamics and fluid mechanics of the DATSS system.

## Competing interests

The authors declare no competing interests.
