## [Peer Review File · Nature Communications]

REVIEWER COMMENTS

Reviewer #1 (Remarks to the Author):

The authors have taken interesting approach for triboelectric/piezoelectric based sensor by using it to detect stall condition for aircrafts. The authors have emphasized the originality of the device and considered mechanical aspect as well for using it to extreme condition aircrafts. The experiments and analysis were precise and manuscript is written well. I have few comments and questions for the authors before I can recommend this manuscript to be accepted.

- As the sensor is based on fluttering motion, it would naturally decrease the efficiency of the aircraft by introducing new friction and drag to the wings. I agree with the authors that the current device is very lightweight, but disturbance and drag due to installing the fluttering sensor should not be considered lightly. The authors should specify the limitation of current device and how the fluttering device would affect the performance of the wings in the discussion section.

- Although the authors have shown meaningful correlation between the stalling condition and electrical output of the sensor, the reason why it produces voltage signals in certain conditions seems to be lacking. It would be clear if the authors provide schematics or photos of DATSS connected with the raw signals.

Reviewer #2 (Remarks to the Author):

The authors found and well demonstrated an interesting application of triboelectric/piezoelectric sensors with simple, flexible and lightweight structural features on detecting surface turbulence status of aircraft wings for flight safety. Triboelectric and piezoelectric sensing materials/structures were elaborately designed and integrated into one flexible strip-like device that can be conformably mounted on the wing of aircraft. It worked to generate triboelectric signals via the repetitive taps between elastic steel sheet and silk fibroin and piezoelectric signals by bending silver-coated PVDF electrode. Based on recirculation wind tunnel trials and data processing, they were demonstrated to be well suitable for monitoring the two states of aircraft, namely the normal flight and the stall state, respectively. The authors also showcased the industrial potential of this stall sensing system by developing an automatic wireless modular and conducting real-world field tests. The unique application-oriented experimental design, well expected results and reasonable data interpretation

suggest that this work is of significance and shall be published in Nature Communications. However, the concerns below shall be addressed before publication.

(1) In Fig. 3, the duration time of stall state indicated by T-Signal disappearing (Fig. 3a and 3b) seems not to be consistent with that indicated by P-Signal (Fig. 3d and 3e), why? The T-signal gradually disappearing process is important for early warning. This signal shall be investigated and analyzed more systematically by considering different flight conditions, such as AOA and velocity.

(2) The P-Signal seems to be generated suddenly during the transition process from normal flight to stall state? How about the piezoelectric sensitivity? Because only piezoelectric sensing element works during the stall state, to “visually” monitor this state, it’s better to investigate and discuss the influence of vibration frequency on the P-Signal under different bending angles.

(3) An interesting adhesion problem using negative triboelectric material was stated in the manuscript. And the author ascribed its main reason to the rapidly accumulated negative charges. Any other reasons? Such as the quantity of accumulated charge? The modified CSFE avoids this adhesion problem, how about other positive triboelectric material, and how about other negative triboelectric material with less charge density? The accumulating process shall be discussed by investigating the triboelectric charge with time/tapping times.

(4) Is T-signal strong under extreme flight conditions, such as freezing and raining conditions? Can it be distinguished explicitly for monitoring the pre-stall process to give warning early?

Reviewer #3 (Remarks to the Author):

Digital mapping of surface turbulence status and aerodynamic stall on wings of a flying aircraft

The authors reported a non-invasive and lightweight active system that can sense and warn the pre-stall and during stall of fixed-wing aircraft by employing conjunct signals are as provided by triboelectric and piezoelectric effects.

- What are the noteworthy results?

I couldn't find the noteworthy results. I expected that the devices applied to the realistic airfoil geometry of the commercial aircrafts and the experiments were conducted at realistic flight Reynolds numbers.

- Will the work be of significance to the field and related fields? How does it compare to the established literature? If the work is not original, please provide relevant references.

I don't think that the work will be of significance to the field and related fields. I didn't find that the authors compared the results with those of other methods.

- Does the work support the conclusions and claims, or is additional evidence needed?

I don't think that the authors showed sufficient results. It is necessary to show additional evidence that the system works other conditions (i.e., different Reynolds numbers, airfoil geometries, and others). In addition, the authors might consider following points:

- Please calculate the angle of attack of the aircraft in the field test and discuss the results of sensing.
- Please provide the total cost of the system developed in this study.
- Please study the accuracy of the sensing in detail.
- Please discuss the measuring limits and the application limits.
- Is it possible to sensing the flow separation associated with shock waves? Please comment on this regard.

- Are there any flaws in the data analysis, interpretation and conclusions? - Do these prohibit publication or require revision?

Effects of the device attached on the wing on aerodynamic performance might be required. Especially, in the case of realistic flight Reynolds number flow of the commercial aircrafts, the boundary layer becomes thin and is expected to be affected by the existence of the devices.

- Is the methodology sound? Does the work meet the expected standards in your field?

The methodologies of experiments look okay. But I don't think that 2D RANS are suitable for these cases considered in this paper and the accuracy of sensing and measuring limits should be presented.

- Is there enough detail provided in the methods for the work to be reproduced?

Turbulent property at the inlet flow of CFD is missing.

Point-to-Point Responses to the Referee**REVIEWER COMMENTS**

Reviewer #1 (Remarks to the Author):

The authors have taken interesting approach for triboelectric/piezoelectric based sensor by using it to detect stall condition for aircrafts. The authors have emphasized the originality of the device and considered mechanical aspect as well for using it to extreme condition aircrafts. The experiments and analysis were precise and manuscript is written well. I have few comments and questions for the authors before I can recommend this manuscript to be accepted.

Response to Reviewer 1: We greatly appreciate the time and valuable comments of the reviewer.

As the sensor is based on fluttering motion, it would naturally decrease the efficiency of the aircraft by introducing new friction and drag to the wings. I agree with the authors that the current device is very lightweight, but disturbance and drag due to installing the fluttering sensor should not be considered lightly. The authors should specify the limitation of current device and how the fluttering device would affect the performance of the wings in the discussion section.

Response: Thank you very much for your comment. We highly agree with the points you made: as the sensor is based on fluttering motion, it would naturally affect the efficiency of the aircraft by affecting the friction and drag of the wings. When designing the DATSS system, we have fully considered the impact on the flight efficiency of the aircraft. For the influence of the flight efficiency of the aircraft, two factors are mainly considered, lift and drag.

(1) For drag, the introduction of DATSS system mainly affects friction resistance and interference resistance. **The DATSS system should minimize the effects of drag.** Firstly, the interference resistance is discussed. DATSS adopts a streamlined design in configuration. The ultra-thin hollow alloy steel sheet can conform to the airfoil of the aircraft and minimize the generation of interference resistance. When the T-signal (triboelectric signal) is working, we optimized the length of the steel sheet to make the vibration amplitude within 1 cm (as shown in the Supplementary Video 6, different airfoils can be optimized differently), the height (1 cm) is less than the height of some

screws on the airfoil of commercial aircraft. The counter electrode is also thin to produce less interference resistance. Next, we discuss the friction resistance. When designing the DATSS system, in order to reduce the frictional resistance, we selected a **stainless steel material with a smooth surface** (low friction coefficient), and designed a hollow structure to reduce the surface area and mass of the DATSS system. The quality of the entire hollow alloy steel sheet is 0.9 g. In addition, the stiffness of the steel sheet is low, and its natural state is a streamlined structure that fits the airfoil. This makes it aerodynamically better for airflow over the surface. We have added these analyses in the Supplementary Information of revised manuscript.

(2) For lift, **DATSS system acts like active vortex generator, that can slightly delay airflow boundary separation and reduce pressure drag.** When designing the DATSS system, we referenced a large number of aircraft design literature, from which we conceived a reasonable system structure that can increase a part of the lift (which may have very little impact, we explain from the theoretical side). i) Blocking effect of reversed air, when the airfoil flow separation starts, the rolled steel sheet can use its rolled-up tail to block the reversely separated airflow while generating the P-signal (piezoelectric signal), so that this part of the airflow can be redirected back to the incoming flow direction of the aircraft. This blocking effect can play a part in delaying the stall by reducing pressure drag. ii) Recovery airflow effect. In the early stage of airflow separation, the steel sheet that flutters up and down will exert downward force on the airflow below during the descending process, so that this part of the airflow delays the airflow separation from the airfoil (similar to the action of a vortex generator)^{1,2}. We have added the above experimental conclusions to the Supplementary Information of the revised manuscript, highlighted in yellow.

(3) **Experimental supplement.** To verify the above theory, we supplement the relevant experiments of the lift test. The lift force of the model was tested using a six-axis force balance in a recirculation wind tunnel. Six-axis force balance is a standard equipment used to test lift force of aircraft. The device is shown in **Fig. R1**, considering that gravity might cause some interference with the lift test, we installed the model wing perpendicular to the direction in which it operates, so that gravity and lift are perpendicular to each other without interference. We compared and tested the airfoil lift with four DATSS units and without DATSS, as shown in **Fig. R2**. It can be seen from the figure that even if the DATSS system is installed on the whole airfoil in array, it has almost no impact on the

airfoil lift, and even more than the original lift at some data points (in the figure, the wind speed in the wind tunnel increased gradually over time, finally, the final lift value was compared).

Fig. R1 Photograph of the wind tunnel test of the six-axis force balance.

Fig. R2 The airfoil lift with four DATSS units and without DATSS.

(4) **CFD simulation.** In addition to the analysis and discussion from the experimental point of view, we also considered the influence of the existence of the sheet on the lift and drag of the aircraft wing from the point of view of CFD simulation. Here, we consider the most extreme case, that is, to study how much lift and drag changes the steel sheet can bring to the airfoil without considering the aforementioned blocking backflow and downpressure airflow (both of which are beneficial for lift). At the same time, in order to demonstrate the arrayable installation of the DATSS system, we placed four steel sheets on the wing section for simulation.

i) CFD parameter setting details: the incoming flow velocity is 200 m/s, the chord length is 0.3 m, and the span length is 0.5 m. Reynolds number: 4,107,522 (which can be identified as a high Reynolds number problem). Wing: NACA0012. Here we analyze the influence of the installation of the steel sheet on the lift drag of the airfoil. Since NACA0012 is a symmetrical wing, the lift of the symmetrical airfoil is basically close to 0 at an angle of attack (AoA) of 0° , which has no practical research significance. Therefore, we choose three cases where the AoA is 6° , 8° , 10° for CFD simulation. The software we used is ANSYS2020-R2.

ii) Description of simulation mesh convergence: when there is no steel sheet on the airfoil, five convergence curves are calculated as residual, lift coefficient, lift, drag coefficient and drag curves as shown in **Fig. R3, R4, R5, R6** and **R7**. The airfoil model without steel sheet has a total mesh number of 2 million.

Fig. R3 CFD simulation convergence calculation.

Fig. R4 Lift coefficient curve without steel sheet.

Fig. R5 Lift curve without steel sheet.

Fig. R6 Drag coefficient curve without steel sheet.

Fig. R7 Drag curve without steel sheet.

iii) The convergence curves, lift coefficient, lift, drag coefficient and drag curves with steel sheet on the airfoil are shown in Fig. R8, R9, R10, R11 and R12 (because the number of meshes of the 3D model in this simulation is huge and the trial calculation time is long, here we only show the first 2800 steps of the curve of the trial calculation, and the trend can be seen from the curve, and the subsequent specific calculations have been completed on the supercomputer). The total number of grids of the airfoil model with steel sheets is 10.55 million, and the height of the first layer of grids is 0.002 mm.

Fig. R8 The convergence curve with steel sheet.

Fig. R9 Lift coefficient curve with steel sheet.

Fig. R10 Lift curve with steel sheet.

Fig. R11 Drag coefficient curve with steel sheet.

Fig. R12 Drag curve with steel sheet.

iv) ICEM grid and fluent settings: the unstructured grid is used for division, and the size of the external flow field is 10 times the chord length. When meshing, the leading and trailing edges of the wing and the tip of the wing are locally refined. At the same time, the steel sheet needs to be encrypted for the model. Since the thickness of the steel sheet is ultra-thin and the distance to the upper airfoil is relatively close, the grid needs to be dense enough to ensure the restoration accuracy of the model. Boundary layers can also create problems that are difficult to generate because they are too close together. Therefore, we use the operation of making a boundary layer first, and then segmenting it to realize the generation of the boundary layer mesh. The grid diagram is shown in **Fig. R13, R14, R15, R16 and R17**. We can observe the position of the encrypted part and the boundary layer. According to the analysis of the calculation conditions, it is a high Reynolds number problem. It is roughly estimated that the grid height of the first layer is 0.002 mm, and there are 15 layers in total.

Fig. R13 Schematic diagram of CFD model grid distribution.

Fig. R14 Schematic diagram of CFD wing front grid distribution.

Fig. R15 Schematic diagram of DATSS systems grid distribution.

Fig. R16 Side view of CFD modeling grid distribution.

Fig. R17 A local diagram of the DATSS system modeling subdivision grid.

v) Software parameter setting (ANSYS2020-R2):

No.	Set	Sort	Parameter
1	Solver	Type	Pressure-Based
		Velocity formulation	Absolute
		Time	Steady
2	Models	Energy	On
		Viscous	SST-K- ω
3	Materials	Air	Ideal-gas
		Viscosity	Sutherland
4	Operating Conditions	Operation Pressure	101,325 Pa
5	Boundary Conditions	Mach Number	0.5877270830
		X-Component of Flow Direction	6°, 8°, 10°
		Y-Component of Flow Direction	6°, 8°, 10°
		Turbulence Intensity	5%
		Turbulent Viscosity Ratio	10
6	Solution Methods	Scheme	SIMPLEC
		Skewness Correction	0
		Gradient	Least Squares Cell Based
		Pressure	Second Order
		Density	Second Order Upwind
		Momentum	Second Order Upwind
		Turbulent Kinetic Energy	First Order Upwind
		Specific Dissipation Rate	First Order Upwind

vi) Calculation results:

AoA	No steel sheet installed				Steel sheet installed				lift coefficient (C _l) ratio	drag coefficient (C _d) ratio
	C _{l-1}	L ₁	C _{d-1}	D ₁	C _{l-2}	L ₂	C _{d-2}	D ₂	C _{l-1} /C _{l-2}	C _{d-1} /C _{d-2}
6°	0.401656	1517.2147	0.031661	119.59552	0.401226	1515.5920	0.031389	118.5695	1.001071715	1.008665456
8°	0.533488	2015.19929	0.0467828	176.71777	0.533979	2017.0561	0.0465215	175.7056	0.999080488	1.005616758
10°	0.650746	2458.13043	0.0669403	252.860878	0.641714	2424.0139	0.0659274	249.0345	1.014074806	1.015363870

The unit of force is N. The convergence trend of small AoA is stable, and the monitoring data is basically stable. The maximum residual value of the final calculation is on the order of 1E-3, and the monitoring data changes very slowly, which can be judged as convergent. Through the simulation analysis, it can be seen that the small enough sheet structure has little effect on the lift

before the wing stalls, about 0.1%. If the width of the sheet is further reduced, the effect similar to that of a vortex generator can be achieved. By creating a vortex above the airfoil to accelerate the airflow velocity on the upper airfoil, the effect of increasing lift and reducing drag can be achieved. At the same time, we have analyzed in the preceding paragraphs that the downward pressure air effect of the steel sheet and the blocking backflow effect can have a positive impact on the lift force.

Furthermore, the DATSS system is much lighter and thinner compared with other mounted sensors that are an order of magnitude thicker and much heavier, in a large number of literatures³⁻⁵. We hope that our systematic analysis and experiments above can provide you with a detailed description of the influence of the DATSS system on the lift and drag of the aircraft.

Although the authors have shown meaningful correlation between the stalling condition and electrical output of the sensor, the reason why it produces voltage signals in certain conditions seems to be lacking. It would be clear if the authors provide schematics or photos of DATSS connected with the raw signals.

Response: Thank you very much for your comment. In this work, the digital-visualization array for turbulence stall sensing (DATSS) system works based on the signal of triboelectric (T-signal)/piezoelectric (P-signal) composite voltage sensing signals. For the T-signal, we used the voltage signal of the single-electrode triboelectric nanogenerator as the sensing signal. Its signal generation principle is: when the steel sheet is in contact with the composite silk fibroin electrode (single-electrode mode), the composite silk fibroin electrode is connected to the STM32 microcontroller. Due to the difference in electronegativity between steel sheet and composite silk fibroin electrode, in the process of contact, the surface of steel sheet tends to gain electrons and the surface of composite silk fibroin electrode tends to lose electrons, so electrons rearrange on the two contact interfaces at this time. During the separation process, the surface charges of composite silk fibroin electrode move to form displacement current. The electrical signal is received by the microcontroller and sends to the receiver through WiFi, so as to obtain a continuous triboelectric nanogenerator signal. The original signal is shown in **Fig. R18**, T-signal is the characteristic output signal of single-electrode triboelectric nanogenerator. The authors have more than 10 years' research

foundation in this field since 2012. A self-powered AC voltage signal due to charge transfer is accepted by the electrometer, resulting in a T-signal.

To better understand its signal generation process, we summarized the working principle diagram (**Fig. R19**) of the single-electrode triboelectric nanogenerator of the signal generation process when the DATSS system worked. In this diagram, steel sheet and the silk fibroin surface act as the friction layers, and the copper film (or other conducting electrode) acts as the conduction layer. In the original position, the two friction surfaces are in close contact. Charges are transferred since the different electron affinities of steel sheet and silk fibroin results in net positive charges on silk fibroin and an equal amount of negative ones on the steel sheet surface. Once the two surfaces are separated, the positive charges on the surface of the silk fibroin induce negative charges on the copper film due to the induced electric potential difference between the conduction layer and ground, driving the electrons to flow from the ground electrode to Cu. This electrostatic induction process will continue until nearly all the positive charges on the silk fibroin are screened from the induced negative charges on the copper film. Once the negatively charged steel sheet is close to the silk film, the induced negative charge on the copper electrode is removed, causing free electrons to flow from the copper electrode to the ground until the steel sheet makes contact with the silk film.

Fig. R18 The amplified detail of the T-signal (T-signal interception diagram of Fig. R33 from 1 s to 2 s).

Fig. R19 The working principle diagram of the single-electrode triboelectric nanogenerator of the T-signal generation process.

For P-signal, when the piezoelectric material is subjected to an external force in a certain direction, electric polarization is generated inside, and opposite charges are generated on two surfaces at the same time; when the external force is removed, the material returns to an uncharged state. When the direction of the external force changes, the polarity of the charge also changes; the amount of charge generated by the force on the material is associated with the magnitude of the external force. When the aircraft stalls, the reverse force generated by the airflow separation on the airfoil causes the PVDF piezoelectric sheet to roll up. The greater the reverse force, the stronger the piezoelectric signal. At this time, the signal is collected by the microcontroller and output to the receiver through WiFi. Thank you again for your valuable comments on the DATSS system, which makes this work better accessible to *Nat. Commun.* readers.

References

- 1 Bechert D, *et al.* Biological surfaces and their technological application-laboratory and flight experiments on drag reduction and separation control. *28th Fluid dynamics conference* (1997).
- 2 Kundu PK, Cohen IM, Dowling DR. *Fluid mechanics*. (2015).
- 3 Sadraey MH. *Aircraft design: A systems engineering approach*. (2012).
- 4 Kim Y, Jeon Y-H, Lee D-H. Multi-objective and multidisciplinary design optimization of supersonic fighter wing. *Journal of aircraft* **43**, 817-824 (2006).
- 5 Raymer D. *Aircraft design: a conceptual approach*. (2012).

Reviewer #2 (Remarks to the Author):

The authors found and well demonstrated an interesting application of triboelectric/piezoelectric sensors with simple, flexible and lightweight structural features on detecting surface turbulence status of aircraft wings for flight safety. Triboelectric and piezoelectric sensing materials/structures were elaborately designed and integrated into one flexible strip-like device that can be conformably mounted on the wing of aircraft. It worked to generate triboelectric signals via the repetitive taps between elastic steel sheet and silk fibroin and piezoelectric signals by bending silver-coated PVDF electrode. Based on recirculation wind tunnel trials and data processing, they were demonstrated to be well suitable for monitoring the two states of aircraft, namely the normal flight and the stall state, respectively. The authors also showcased the industrial potential of this stall sensing system by developing an automatic wireless modular and conducting real-world field tests.

The unique application-oriented experimental design, well expected results and reasonable data interpretation suggest that this work is of significance and shall be published in Nature Communications. However, the concerns below shall be addressed before publication.

Response to Reviewer 2: We greatly appreciate the time and valuable comments of the reviewer.

(1) In Fig. 3, the duration time of stall state indicated by T-Signal disappearing (Fig. 3a and 3b) seems not to be consisted with that indicated by P-Signal (Fig. 3d and 3e), why? The T-signal gradually disappearing process is important for early warning. This signal shall be investigated and analyzed more systematically by considering different flight conditions, such as AOA and velocity.

Response: Thank you for your excellent suggestions that helped us to correspond the signal to the stall detection of the aircraft in this paper more specifically and clear. Your comment is very correct, the triboelectric signal (T-signal)/piezoelectric signal (P-signal) in the DATSS system are not synchronized. There is a time difference between the end of the T-signal and the beginning of the P-signal (this time difference does not occur every time and is related to the actual flow field in the test, meanwhile, T/P signals in Fig. 3 of the manuscript were tested by two electrometers, later, we independently developed a signal collection device in Fig. 4 and Fig. 5). When the steel sheet leaves the airfoil surface (at this point, T-signal disappears) due to the initial airflow separation, it will not

bend immediately to generate the P-signal, but when the angle of attack (AoA) continues to increase slightly, after the airflow separation intensifies, the P-signal begins to be generated. The time difference between the T-signal and the P-signal is more obvious in the wind tunnel test, because the AoA of aircraft wing in the wind tunnel changes at a constant speed (Supplementary Video 6). In the field test of the UAV, the time difference between the T-signal and the P-signal is shortened due to the rapid change of the AoA and the uneven air speed. This is also the reason why we use the P-signal to judge the degree of airflow separation after the T-signal senses the airflow separation in this work.

In Fig. 3, the duration time of stall state indicated by T-Signal disappearing (Fig. 3a and 3b) is not to be consisted with that indicated by P-Signal (Fig. 3d and 3e), **the times in Figures 3d and 3e correspond to the 4.0 second to the 8.5 second in Figures 3a and 3b in the original manuscript.** Our initial idea was to visualize the process of the P-signal, so the coordinate axis starting from 0 seconds was used uniformly in the original manuscript. In the revised manuscript, in order to make it easier for reviewers, editors, and readers to correspond to the time relationship, we marked the time axis of Figures 3d and 3e as 4 seconds to 8.5 seconds, so as to correspond to the T-signal.

For the T-signal under different AoA and different wind speeds, we have done a more systematic study, as shown in **Fig. R20**. Your comment is very critical. After further study, we found that there is a relationship between the frequency of T-signal and the speed of wind tunnel motor (wind speed), in other words, in the DATSS system, we can also judge the current wind speed (flight speed) according to the T-signal, which makes the function of the DATSS system has been increased. We have added this finding into the Supplementary Information of revised manuscript. At the same time, we also find that there are many reports on the correlation between the vibration frequency of T-signal and wind speed, but there are few reports under high wind speed. Therefore, this conclusion can supplement the signal characteristics of T-signal under high wind speed. But for the T-signal amplitude, the test uses a Keithley 6514 electrometer for a standard test, and its amplitude increases slightly with wind speed. In addition, we tested the frequency change of T-signal with AoA of 0° to 10° at the same wind speed. It was found that taking motor speed of 1500 rpm as an example, the frequency of T-signal was all around 40 Hz, and no correlation was found.

Fig. R20 The relationship between T-signal frequency/amplitude and fan motor speed.

(2) The P-Signal seems to be generated suddenly during the transition process from normal flight to stall state? How about the piezoelectric sensitivity? Because only piezoelectric sensing element works during the stall state, to “visually” monitor this state, it’s better to investigate and discuss the influence of vibration frequency on the P-Signal under different bending angles.

Response: Thank you very much for your comment. When not stalling, the steel sheet produces T-signal in a nearly flat state, and the P-signal also exists at this time, but it is very weak. When the steel sheet is rolled and generates the P-signal, the signal is much larger than the flat state due to the piezoelectric lattice polarization effect. At the same time, when the air separation force is greater than the component force of the gravity of the steel sheet, the steel sheet will "suddenly" start to roll over and generate a P-signal, so a "sudden" P-signal appears. We tested the sensitivity of the P-signal. Since the piezoelectric effect is more of the effect of force on the voltage signal, the amplitude change of the P-signal at different bending angles is greater than the frequency change.

First of all, we have carried out experimental research on the relationship between the rolling force and the angle of the steel sheet that generates the P-signal as shown in Fig. R21.

Fig. R21 The relationship between the rolling force and the angle of the steel sheet.

It can be seen from the results that only 0.09 N of airflow separation force is needed to generate the P-signal, as the bending angle increases, the required force also increases. This is also the principle that the stall depth can be judged based on the P-signal amplitude. Related supplementary experimental pictures are as follows (Fig. R22).

Fig. R22 Photograph of the test of the relationship between the rolling force and the angle of the steel sheet.

Fig. R23 The P-signal peak value under different bending angles.

At the same time, we measured the P-signal peak value under different bending angles several times, as shown in **Fig. R23**. The stall depth can be roughly judged according to the amplitude of

the P-signal. In future commercial products, the relative change of the P-signal amplitude may be used to analyze the degree of the stall, however, it is necessary to overcome the interference of signal in complex turbulent environment.

Finally, based on your comment, we also explored the effect of frequency variation on the amplitude of the P-signal. Just like your comments, frequency has an effect on the amplitude of P-signal. **Fig. R24, R25 and R26** are the results of measuring the relationship between frequency and amplitude of P-signal with mechanical motor at the bending angle of 30°, 60° and 90° respectively. We test the amplitude change of P-signal at the frequency from 1 Hz to 7 Hz when the bending angle is 30°. It is observed that the amplitude of P-signal tends to increase with the increase of frequency. However, after 4 Hz, this trend gradually slows down and becomes stable. Since the actual operating frequency of the DATSS system is greater than 20 Hz, in summary, the influence of the bending angle on the amplitude of P-signal is greater than that of the frequency. We have added the in-depth analysis of the P-signal to the revised manuscript. Thank you again for your valuable comments.

Fig. R24 The relationship between frequency and amplitude of P-signal with mechanical motor at the bending angle of 30°.

Fig. R25 The relationship between frequency and amplitude of P-signal with mechanical motor at the bending angle of 60°.

Fig. R26 The relationship between frequency and amplitude of P-signal with mechanical motor at the bending angle of 90°.

(3) An interesting adhesion problem using negative triboelectric material was stated in the manuscript. And the author ascribed its main reason to the rapidly accumulated negative charges. Any other reasons? Such as the quantity of accumulated charge? The modified CSFE avoids this adhesion problem, how about other positive triboelectric material, and how about other negative triboelectric material with less charge density? The accumulating process shall be discussed by investigating the triboelectric charge with time/tapping times.

Response: Thank you very much for your comment. The question is a very interesting one, which we also think a lot. As the reviewer commented, the rate and quantity of charge accumulation both affect the electrostatic attraction: the rate affects the temporal attraction over time; the quantity, that will eventually saturate, affects the long-time attraction. We agree with the reviewer that the final quantity of accumulated charge may have more adhesion effect, since the charge accumulation rate in the sensing scenario is rather small. Thus, we revised our manuscript to ascribe the main reason of adhesion to the final quantity of accumulated negative charges.

Since electron will always transfer from one to another material, depending the triboelectric series¹, the attraction will be always there, just a difference in magnitude. Therefore, other materials causing less attraction (other positive triboelectric material and negative triboelectric material with less charge density as the reviewer mentioned) may also work, but may not have the good chemical and mechanical properties as CSFE. Although the attraction effect is experimentally observed in the TENG field in many studies, the effect of electrostatic attraction is rarely systematically studied. In fact, our recent review paper has also discussed the lacking of electrostatic attraction in various models of describing fluttering TENG. The multiphysics coupling of fluid-structure-electrostatic will actually be one of the further research directions for our team².

To investigate the adhesion effect from the final quantity of accumulated negative charges, we have added the surface electrical potential (which corresponds to the degree of electrostatic adhesion force) scanning experiments for CSFE and other materials such as PA (positive), FEP (less negative), and PTFE (more negative). In the experiments, the sheets were automatically scanned from top to bottom, left to right over time using Trek electrostatic voltmeter (model 347) for the surface electrical potential (**Fig. R27**). Two scenarios of the materials, before (**Fig. R28**) and after (**Fig.**

R29) rubbing with the steel sheet, were tested. Materials other than CSFE show unstable electrical potential in their original states (before rubbing) and show much higher (either negative or positive) electrical potential (as high as several thousand V) after a thorough rubbing, which will cause higher adhesion. Among the tested materials, CSFE shows the most stable and lowest electrical potential (<60 V, Fig. R30) before and after rubbing with the steel sheet, minimizing the adhesion problem during sensing while generating stable sensing signals.

Fig. R27 Schematic of the measurement (left) and the automatic scanning route (right).

Fig. R28 Scanned electrical potential of various materials before rubbing with the steel sheet.

Fig. R29 Scanned electrical potential of various materials after rubbing with the steel sheet.

Fig. R30 Scanned electrical potential for CSFE, before and after rubbing.

(4) Is T-signal strong under extreme flight conditions, such as freezing and raining conditions? Can it be distinguished explicitly for monitoring the pre-stall process to give warning early?

Response: Thank you very much for your comment. It is very important to study the influence of complex environment and extreme environment on T-signal, which will be the key factor for the

industrialization of DATSS system. We supplemented T-signal data in the wind tunnel simulating extreme environments, such as freezing and raining conditions. As shown in **Fig. R31** and **R32** (Supplementary Fig. 11 and 12 in original manuscript), in the Supplementary Information of our original manuscript, we have conducted wind tunnel tests for rain and ice conditions. The upper surface of the working area of the wind tunnel, there was an opening area that can simulate rainwater. The freezing mode test is to put the wet wing surface with DATSS system in the refrigerator for 12 h, make the surface freeze, and then put it into the wind tunnel for testing. T-signal data in extreme cases can be used to warn of stall occurrence.

Fig. R31 Stall sensing T-signal of DATSS system under surface icing condition.

Fig. R32 Stall sensing T-signal of DATSS system in rain condition.

Thank you again for your valuable comments on the DATSS system, which makes this work better accessible to *Nat. Commun.* readers.

References

1. H. Zou, Y. Zhang, L. Guo, P. Wang, X. He, G. Dai, H. Zheng, C. Chen, A.C. Wang, C. Xu, Quantifying the triboelectric series, *Nat. Commun.* **10** (2019) 1-9.
2. L.N. Cao, Z. Xu, Z.L. Wang, Application of Triboelectric Nanogenerator in Fluid Dynamics Sensing: Past and Future, *Nanomaterials* **12** (2022) 3261.

Reviewer #3 (Remarks to the Author):

Digital mapping of surface turbulence status and aerodynamic stall on wings of a flying aircraft

The authors reported a non-invasive and lightweight active system that can sense and warn the pre-stall and during stall of fixed-wing aircraft by employing conjunct signals are as provided by triboelectric and piezoelectric effects.

Response to Reviewer 3: We greatly appreciate the time and valuable comments of the reviewer.

What are the noteworthy results?

I couldn't find the noteworthy results. I expected that the devices applied to the realistic airfoil geometry of the commercial aircrafts and the experiments were conducted at realistic flight Reynolds numbers.

Response: Thank you very much for your comment. For the highlights and significance of this work, we made the following summary, and sincerely hope to get your re-evaluation.

(1) We combine active triboelectric and piezoelectric effects to **visually** and **digitally** map the surface turbulence and aerodynamic stall of flying aircraft *in situ*.

Simultaneously realize *in-situ* sensing of digital signals of airfoil separation and visualize the process of air separation, which is something no team can do so far. Today, **the tuft method** is still used for the airfoil airflow boundary layer separation sensing of the existing aircraft in field flight test¹⁻⁵. The significance of ***in-situ* visualization** and **digital monitoring** of airflow separation is that it can not only real-time insight into whether the aircraft is at risk of stall, but also judge the depth of stall. Whether it is for civil aviation or unmanned aerial vehicles (UAV), it is a new stall sensing solution.

(2) From the fundamental of aerodynamics, the arrayed sensing system is rigorously calibrated and validated by the standard recirculation wind tunnel and airfoil model testing, and field measurement, assisted by dynamic simulation using computational fluid dynamics (CFD).

In this work, the digital-visualization array for turbulence stall sensing (DATSS) system achieves accurate sensing results in **wind tunnels**, **UAV field flight test** and **CFD**. These are the three most common and standard test methods for fluid mechanics sensors today. In this work, we gave a lot of effort on sensor accuracy evaluation using all three methods.

(3) The system is optimized in terms of structures and materials targeting broad practical applications, for example, miniaturization, non-invasive, various resistance, and wireless data delivery modular.

From the selection of the experimental direction, to the design of the sensor, to the optimization of the sensor material, and finally to the DATSS system integration, we used the **industrialized technical standard** that can be mass-produced to conduct experiments and tests throughout the process. Such as CNC machining to make hollow alloy steel sheets, large-area flexible silk fibroin electrode technology, STM32 single-chip technology, and WiFi communication technology. This is also the highlight of this work that distinguishes it from many laboratory test-level work.

More importantly, as you commented "I expected that the devices applied to the realistic airfoil geometry of the commercial aircraft and the experiments were conducted at **realistic flight Reynolds numbers.**" **We now conducted extra test for it.** In order to achieve the real Reynolds number level, we made a wing with a large chord length (**600 mm**), and at the same time increased the flow velocity of the wind tunnel (**80 m/s**), keeping the room temperature the same as the outdoor (**0 °C**, increasing the air density), and finally made the **Reynolds number reach 3×10^6** , which is the Reynolds number of real aircraft such as Cessna 172, and is smaller than the Reynolds number of large aircraft such as Boeing 7-series aircraft. Since AoA cannot be continuously changed for 600 mm airfoil, all relevant data and discussion of the supplementary 80 m/s wind speed test experiment for 600 mm airfoil are added to the Supplementary Information of the revised manuscript. This is also a very important revision in order to better study the performance of the DATSS system at real Reynolds numbers. Under the DATSS system with the same design, the test results are shown in **Fig. R33** (using the same test single chip microcomputer as the flight in the field). We showed the original data of the signal, from which we can analyze: i) the amplitude of T-signal increases after the Reynolds number increases. ii) As the Reynolds number increases, the

noise signal increases. iii) P-signal increases slightly after Reynolds number increases. iv) It can be seen from the signal shape that the real Reynolds number airflow field model changes little, which is consistent with the speculation of self-similarity region. At the same time, it is worth noting that the signal trend under the real Reynolds number in the wind tunnel is the same as that under the field flight, but different in amplitude.

Fig. R33 The T/P-signal test of DATSS system at the Reynolds number reach 3×10^6 .

At the same time, we referred to the test Reynolds number of the world's advanced aircraft wind tunnel platform, as shown in **Fig. R34**⁶, its order of magnitude is comparable to the wind tunnel used in our work. As for the flight of the real aircraft, we have contacted the relevant flight companies and informed that the components to be installed on the outside of the aircraft, regardless of their size, must be professionally authenticated by the civil aviation administration to obtain the airworthiness component number before installation. This requires more in-depth research and engineering work in the future.

Fig. R34 Transonic wind tunnel operating envelope comparison⁶.

The existing real Reynolds number wind tunnel test and UAV field flight test can support the feasibility and accuracy of the DATSS system. Through a large number of preliminary investigation and design improvement, we elaborated from the following three aspects, (1) approximate flow field, (2) a large number of references (3) stall angle deviation to a large angle.

(1) Approximate flow field. The approximate problem of flow field will be discussed in detail in the following answer to the question of system accuracy. We chose the Reynolds order of 10^5 on the one hand because of a large number of relevant literature reports, and on the other hand to study stall warning stability at low Reynolds number. With the increase of Reynolds number, the flow enters the completely turbulent rough zone, and the flow energy loss is mainly determined by the pulsation, and the influence of viscosity can be ignored. At this time, as the Reynolds number continues to increase, the disturbance degree and velocity profile of the fluid in the wind tunnel almost do not change, and the loss coefficient along the wind tunnel does not change, but is only related to the relative roughness. Reynolds criterion has lost its role in distinguishing similarity. This state is called the self-modularization state, and the Reynolds number range of the self-

similarity state is called the self similarity region ($Re \sim 10^5 - 10^7$). So, a Reynolds number of 10^5 can simulate the flight scenario of an ordinary aircraft.

(2) A large number of references. A large number of peer wind tunnel experiments have been reported, so that we have sufficient theoretical reference when designing experiments. At the same time, the recirculation wind tunnel we used in this work is in a relatively advanced level. The Reynolds number of wind tunnel flow field used in wind tunnel experiments of relevant aircraft is listed in the following table.

Ref.	Airfoil	Re
7	NACA 0018	1.5×10^5 to 1×10^6
8	NACA 0018	0.6×10^5 to 1.4×10^5
9	NACA0012, SG6043, SD7062, DU06-W-200	0.65×10^5 to 1.5×10^5
10	NACA 1412, NACA 2412 , NACA 4412	1×10^5 and 2×10^5
11	NACA 2412	1×10^5 to 5×10^5
12	S1223	1×10^5 and 1.35×10^5
13	NACA 0012	3×10^5 and 5×10^5
14	NACA 0015	1.67×10^5 to 5×10^5
15	FX63-137, M06-13-128, S1223	2×10^5

(3) Stall angle deviation to a large angle. As we know, with the increase of Reynolds number, the critical angle of stall for the same airfoil will increase. In other words, the reliability of early warning can be guaranteed as long as the experimental Reynolds number is designed under the lower Reynolds number of this airfoil. Because the warning logic is valid for higher speeds after this Reynolds number.

Thank you again for your comment which enables us to do this work at the larger Reynolds number, which is crucial for the improvement of this work.

Will the work be of significance to the field and related fields? How does it compare to the established literature? If the work is not original, please provide relevant references.

I don't think that the work will be of significance to the field and related fields. I didn't find that the authors compared the results with those of other methods.

Response: Thank you very much for your comment. It is very necessary to compare the existing technology to reflect the superiority of the DATSS system. Before designing the DATSS system, we have reviewed a lot of literature and compared a large number of systems in the field of stall sensing. Now we elaborate as follows.

First of all, we confirm that the DATSS system based on triboelectric (T-signal)/piezoelectric (P-signal) composite voltage sensing signals is the **original work** of all the authors of this paper, and the idea and experimental plan come from the corresponding author, Professor Zhong Lin Wang. Before this, there is no relevant report on the application of triboelectric/piezoelectric composite signal to aircraft stall sensing.

For the existing stall sensing systems, we conducted a comparative analysis:

(1) **The tuft method.** Nowadays, the stall visualization sensor device used in large numbers in the field flight test of aircraft is still the tuft method that has existed for a long time (as shown in **Fig. R35**, www.youtube.com/watch?v=C3xE07kauos&t=416s), here, all pictures on websites are attached with the links.

Tuft method for various types of aircraft

Fig. R35 The tuft method for stall sensing of aircraft.

Besides, the tuft method is also widely used in wind tunnel testing, as shown in **Fig. R36** (www.youtube.com/watch?v=cQ9igGdd-EQ www.youtube.com/watch?v=-xRw_zsSd1k&t=18s).

Fig. R36 The tuft method for wind tunnel test of aircraft wings.

Besides, many automotive aerodynamic tests also use the tuft method. Through the visual swing of the tuft array during high-speed driving, the direction of the surface flow field can be judged, so as to design a vehicle with low wind resistance and high downforce (**Fig. R37** www.youtube.com/watch?v=m8_Z_vskvrs&t=438s/www.youtube.com/watch?v=muXKoOMWD0Y&t=1091s).

Fig. R37 The tuft method for automotive aerodynamic testings.

In addition, there is a lot of literature on the tuft method and has a large time span. It is the most mature and most used method for visualizing turbulence at present. In this article¹⁶, the author introduces a flow visualization method for detecting airfoil separation by a **special tuft method**. This method is the first widely used flow separation detection method, the simplest and the most direct. The experimental results of fluorescent microfilaments and flow cones are presented in this paper. The fluorescent microfilament method uses a thin thread with a diameter of less than 0.006 inches and a length of 1 inch treated with fluorescent dyes. It is irradiated with strong ultraviolet rays at night, so that the observer can directly observe the air flow state of the airfoil and propeller surface during flight. Flow cones solve the problem of whiplash effect of thin lines due to their own material conditions, and can easily distinguish airflow separation. The advantage of the tuft method also contains that it can monitor the air flow separation in an array. However, its **disadvantage** is also obvious. Without a digital signal, it is impossible to **well document and precisely judge** the degree of air flow separation.

(2) **Smoke flow method**. The smoke flow method is commonly used for flow visualization in wind tunnels as shown in **Fig. R38** (www.youtube.com/watch?v=Png0fnG0b_U&t=48s www.youtube.com/watch?v=EfEFjFT8gFU www.youtube.com/watch?v=rCpZpKZLz14). It visualizes the flow of air by evaporating a liquid that can generate smoke from a filament by electrically heating. It is a common flow visualization method in laboratories. Under the smoke, the flow of the airflow can be seen with the naked eye, the development of the airflow over the airfoil surface can be observed, and the separated shear layers can be studied in detail. The disadvantage of the smoke flow method is that it can only observe the low speed of the incoming flow. At the same time, to refine the trajectory of the smoke, it requires high equipment costs and is difficult to use in field flight tests. Therefore, the smoke flow method can only be used in the aircraft design stage to provide the aircraft designer with a reference for the airflow separation under different shapes. Strictly speaking, the smoke flow method is only a support tool for the wind tunnel device, not a system that can be mounted on the aircraft¹⁷.

Fig. R38 The smoke flow method for wind tunnel test of aircraft wings.

(3) **Angle of attack (AoA) sensing system.** AoA sensing system is the most common solution for early warning of stalls in existing civil aviation aircraft. It monitors the aircraft's AoA in real time through a variety of sensors, and indirectly warns the occurrence of stalls. Many literatures also summarize the AoA sensing method in detail. In this article, the authors introduced a sensing system for measuring the AoA of a small aircraft¹⁸. The AoA porous probe and estimation method are used to measure the AoA of the aircraft in real time. When the AoA is too large, it will give an early warning of stall. However, this system requires enough space to install probes and hardware, and the structure is complex, which cannot be applied in the field of UAV. Similar AoA sensing systems also have many kinds of structures. The main disadvantage is that the structure is complex, and it needs complex algorithm support to indirectly warn the occurrence of stall. Air crashes due to faulty AoA sensing systems, like the two Boeing 737-MAX8 accidents, are not far off.

(4) **Other non-industrial airfoil turbulence sensing systems.** In order to solve the problems of the existing stall sensors mentioned above, many novel stall sensing literatures are emerging. i) Huang et al. reported a multifunctional aircraft intelligent skinning system that can simultaneously monitor temperature, stress, and vibration¹⁹. The device is attached to the surface of the wing to form a sensing skin. The maximum thickness of the sheet is about 80 μm , in contrast, our DATSS device

is thicker and the boundary layer effect is smaller than that of Huang et al., the output signal of the sensor is recorded and encoded by an analog-to-digital converter, and finally transmitted to the data processing center. These data were used to identify flight conditions and structural states. The above-mentioned intelligent sensing system has multiple functions and environmental adaptability, but has a complex structure and requires complex algorithms and is not actively signaling. ii) Jiang et al. fabricated artificial whisker sensors by 3D printing and (directional liquid spreading, DLS) technology²⁰. The sensor can sense complex flow details, including flow velocity, vibrations caused by eddy currents, and oscillating flow. The sensor can sense subtle flow and capture the details of the flow, but it cannot directly sense the flow state of the airfoil from the appearance state, and requires more complex back-end calculations.

We will write the above comparison into the revised manuscript, so that readers can compare the performance and principle differences of different stall sensing devices more intuitively, and then reflect the originality and innovation of the DATSS system: simplified structure, active signal transmission, visualization and digitization integration, arrayable design, early warning of stall and stall degree, lightweight and miniaturized system, simple algorithm and low cost. For the aircraft industry, the DATSS system can be used in wind tunnels during aircraft design and development, and can be used with airfoils when flying in the field.

Does the work support the conclusions and claims, or is additional evidence needed?

I don't think that the authors showed sufficient results. It is necessary to show additional evidence that the system works other conditions (i.e., different Reynolds numbers, airfoil geometries, and others). In addition, the authors might consider following points:

- Please calculate the angle of attack of the aircraft in the field test and discuss the results of sensing.
- Please provide the total cost of the system developed in this study.
- Please study the accuracy of the sensing in detail.
- Please discuss the measuring limits and the application limits.

• Is it possible to sensing the flow separation associated with shock waves? Please comment on this regard.

Response: Thank you very much for your comment. The data for different Reynolds numbers, airfoil geometries were listed in the supporting information in the revised manuscript, thank you for your valuable comments.

Due to the lack of monitoring conditions, the AoA of the field flight experiment can only be roughly calculated from the photos in the previous experiments, which is around 18-20°. The AoA measurement in the wind tunnel can be calculated in real time, but most of the UAVs used for filed flight monitoring are not equipped with real-time AoA sensing systems.

For this field flight experiment, our original intention is to prove that it has highly recognizable T/P-signal in the real environmental flow field to sense the airflow separation on the surface of the airfoil. The DATSS system is not an AoA sensing system, upon request, DATSS system is a stall sensor that monitors airflow separation. **For its industrial applications, it can be customized according to the needs of its application scenarios.** For example, the aircraft needs to warn the airflow separation signal at the beginning of the inflection point of the lift coefficient, and the steel sheet stiffness of the DATSS system can be reduced to allow a small separation air force to generate a P-signal; on the contrary, the aircraft needs to warn the airflow separation signal at the moment of the lift coefficient drops to close to 0, we can increase the rigidity of the steel sheet, so that a larger separation air force can generate a P-signal. **This is the advantage that the DATSS system can adjust the timing of the early warning according to the needs of various airfoils and Reynolds numbers, which is also what the AoA sensor cannot do.** We'll discuss the relationship between DATSS system sensing and the AoA of airfoil in detail which concerns the accuracy issue you commented on.

We have made an estimate of the total cost of the entire DATSS system as follows, and we know that after a system is commercialized, its total cost can be greatly reduced.

1. The design and modeling of the hollow alloy steel sheet is done by the authors, and the CNC processing fee is \$2.79 per sheet (according to the real-time exchange rate at 3:30 pm on November 26, 2022, the same below).
2. The price of PVDF piezoelectric film is \$167.36 per sheet (A4 size), and the average price of each DATSS unit is \$1.14
3. The design of the flexible PCB is done by the authors, and the processing fee is \$5.10 each
4. The composite silk fibroin film is designed, modified, prepared and optimized by the author's team, and the material cost is \$10.50 per piece
5. The single-chip system (including the WiFi transmission and reception terminal) is designed and optimized by the authors, and the cost of the whole machine is \$70.42
6. The LabVIEW system is developed by the authors themselves.
7. The material cost for the circuit part is \$1.12.
8. Wind tunnel test and CFD are provided from the project platform of the authors.

The total cost for one DATSS system is approximately **\$91.07**. We have confirmed the source of the fund in the Acknowledgments section.

According to your valuable comments and our design logic when developing the DATSS system, we have studied the accuracy of the sensing in detail and this part will also be highlighted as an important revised part of the revised manuscript.

First, we discuss the accuracy from the principle of the DATSS system, the principle of the stall warning of DATSS is *in situ* monitoring of the separation of airflow, which is essentially different from the AoA sensor. AoA sensors are indirectly monitored for stalls, when AoA is greater than the preset warning value, it will send out a warning to the pilot or drone controller, which may cause a stall. (It is worth noting that the logic we are talking about here is that the aircraft may be stall because the AoA sensor cannot directly accurately judge the stall.) Stall is connected with

different flight speeds, different aircraft wing parameters, and different flight environments. At this time, the AoA sensor can only play a warning role. Whether the stall is occurred requires the operator to judge independently. Therefore, based on the above facts, the DATSS system uses the airflow separation to determine the occurrence and depth of the stall, in principle, it is better than the AoA sensor.

Next, we discuss the accuracy of early warning of the DATSS system. Before answering this question, and at the same time before we design the DATSS system, we have already discussed the stall warning logic of the DATSS system, which stage of the stall needs to be warned, or when does the warning start? We know that it is inaccurate to simply monitor the aircraft AoA to warn of the stall. As the Reynolds number increases, the stall AoA will also increase (this increase is small), so the DATSS system chooses to use airflow separation monitoring to warn of a stall. The broad definition of stall AoA is the AoA when the lift coefficient of the aircraft begins to decrease. As the AoA continues to increase, the lift coefficient decreases at an accelerated rate, eventually leading to a crash. We first calculated and simulated the lift coefficient curve of our wind tunnel experiment airfoil NACA0012, as shown in **Fig. R39**. When the AoA is about 16° , the lift coefficient begins to decrease (the experimental simulation Reynolds number is about 600,000). For different software and different references, the calculation method of the lift coefficient is slightly different, so it is necessary to slightly adjust the early warning angle according to the measurement when it is commercially available. We tested the angle of attack of the DATSS system at P-signal generation 100 times of the original manuscript, and the results all fell around 16.2 degrees. Since its principle is based on the sensing of airflow separation, its accuracy is higher than that of the AoA stall sensing scheme. In addition, we also calculated the drag coefficient and lift-to-drag ratio changes of the NACA0012 airfoil under different AoAs and different Reynolds numbers, which can be used as an experimental reference, as shown in **Fig. R40** and **R41**.

Finally, we discuss the issue of Reynolds number. In wind tunnel experiments, when the Reynolds number is large, the turbulent eddy viscosity dominates, and the molecular viscosity is almost negligible. Therefore, as long as the Reynolds number is larger than a certain value, the flow is similar by default. This value is the value of the fully developed turbulent flow, which is generally considered to be 10^5 - 10^7 . Therefore, as long as the Reynolds numbers of the real flow and the model

experiment are large enough, they are considered to be similar, and this Reynolds number range is usually called the self similarity region. Therefore, we have set the Reynolds number to 10^5 in the design phase of the wind tunnel experiment. At this time, the boundary layer can be regarded as full of turbulent flow, and the influence of the Reynolds number (10^5 - 10^7) on the flow can be ignored. In this Reynolds number range, when other conditions remain unchanged, the airflow separation point basically remains unchanged. Most commercial aircraft have a Reynolds number of 10^6 , so our model is in a similar range to the **real aircraft Reynolds number**.

Fig. R39 The lift coefficient curve of experiment airfoil NACA0012.

Fig. R40 The drag coefficient curve of experiment airfoil NACA0012.

Fig. R41 The lift-to-drag ratio changes curve of experiment airfoil NACA0012.

Here, it is worth noting that after answering the accuracy question, we would like to share the "customizable" features of the DATSS system as shown in Fig. R42. The customization of the DATSS system is reflected in three aspects:

Fig. R42 The lift coefficient curve of NACA0012.

(1) **Adjustability.** As shown by the black arrow in Fig. R42, firstly, the stall is not a flight state corresponding to a specific AoA, and the stall is a whole range after the lift coefficient drops until the lift is lost. Different from existing AoA stall sensors, it can only warn the stall after a specific AoA. The early warning of the DATSS system can be artificially set at most locations in the entire stall area, which is in line with the early warning logic (it is not difficult to understand, only need to adjust the stiffness, thickness and length of the steel sheet, early warning can be given at the position where warning is needed). What are the benefits of this? The logic is that some pilots or UAV operators prefer to receive an early warning signal when the airflow separation begins, so as to ensure the flight safety of the aircraft to the greatest extent; on the contrary, other controller

prefer the joy of controlling the aircraft with a large AoA, hoping to receive a stall warning when the airflow separation is serious. In this way, the DATSS system with adjustable warning position is undoubtedly more advantageous.

(2) **Safety Threshold.** As shown by the **red arrow** in **Fig. R42**, as the Reynolds number (Re) increases, the critical stall AoA of the aircraft will also increase. Therefore, in the wind tunnel test, we use a Reynolds number on the order of 10^5 (when $Re > 10^5$, the wind tunnel can simulate the flow field with Re between 10^5 and 10^7 as the fluid model is similar) to ensure the safety critical threshold of the DATSS system. That is to say, at a small Reynolds number, the DATSS system can work normally in early warning, as the Reynolds number increases, the critical stall AoA of aircraft will only become larger, and no less than a low Reynolds number.

(3) **Self Similarity Region Reliability.** As mentioned in the answer to the real Reynolds number problem, in this work, it is not necessary to consider the influence of Mach number in low speed incompressible flow, and the flow state is mainly affected by Reynolds number. The various properties of incompressible flows are generally described as varying with the Reynolds number, and as long as the Reynolds number is approximate equivalence, it is usually ensured that the conclusions of model experiments can be applied to real flows. If the real flow field is too small to be measured (for example, the lift force of insect wings), a larger model can be used to replace it in the experiment, and the velocity can be reduced appropriately to ensure the approximate equivalence of Reynolds number. If, on the other hand, the actual flow field is too large to be achieved in a wind tunnel (such as studying the aerodynamic forces on a skyscraper), the size needs to be reduced and the velocity increased. In addition, many natural air flows are often tested in water tunnels, where the Reynolds number is approximate equivalence and the effect of gravity is excluded. **This idea is fine in theory, but difficult to implement.** For example, a skyscraper with a height of 400 m is limited by the size of the wind tunnel. If the model with a height of 0.4 m is used instead in the experiment, the wind speed in the wind tunnel needs to be 1000 times of the actual wind speed in order to ensure the same Reynolds number. If the actual wind speed is 10 m/s, then the wind speed in the wind tunnel should be 10 km/s. Regardless of whether such wind speeds can be achieved in a wind tunnel, the point is that this is already hypersonic flow, and compressibility and changes in the physical properties of the air make the flow completely different. Therefore, even for

incompressible flows, the Reynolds number of the model experiment is often not equal to that of the actual flow. How do we make sure that the flow is similar (approximate equivalence)? When the Reynolds number is large, the turbulence vortex viscosity takes the dominant position and the molecular viscosity can be almost ignored. Therefore, the flow is almost similar as long as the Reynolds number is large to a certain value, which is the value that guarantees the flow to be fully developed turbulence, generally believed to be $10^5 \sim 10^7$. Therefore, as long as the Reynolds numbers of the real flow and the model experiment are large enough, they are considered to be similar, and this range of Reynolds numbers is often referred to as the **self-similarity region**. Numerous wind tunnel experiments are also based on this. The airflow separation flow field we focus on in this work can be regarded as equivalent to the real flow field.

Thank you for your comment on whether there are some limitations of the DATSS system that were less analyzed in the original manuscript. We discuss in detail here. Since the vibration and rollover of the steel sheet need to be stressed (already analyzed in the original manuscript), the minimum force to support its vibration and rollover is the limit of the working conditions of the DATSS system. After experiments, we found that when the incoming flow velocity in the wind tunnel is less than 15 m/s, no matter how the AoA changes, the T/P-signal cannot be generated. Therefore, the working speed greater than 15 m/s is the limitation of the DATSS system for the minimum working speed (in general, civil aviation aircraft and most drones can meet this speed requirement). At the same time, whether the DATSS system can work normally at an ultra-high Mach number, we cannot get a clear answer under the current experimental conditions. We have carried out project cooperation with an ultra-high Mach number wind tunnel research institution, which will be the focus of our research in the future. Finally, the density of the air will also affect the signal of the DATSS system. According to the literature²¹, the T-signal will produce a larger signal output in a vacuum environment, that is to say, in the flight environment with lower air density, the T-signal will change.

For the discussion of shock waves, first of all thank you very much for your comments, this is a very challenging and important area. For the generation of shock waves, its flight speed must be greater than 340 m/s, that is, to fly at a high Mach number. As we mentioned above, our experimental research on airflow separation at ultra-high Mach numbers is still in the preparatory

stage. We have purchased a Schlieren airflow imaging system and cooperated with high Mach number wind tunnel institutions to carry out research in this direction. At the same time, from a theoretical analysis point of view, the DATSS system can still work normally in a high Mach number flight environment that can generate shock waves. After the shock wave is generated at high speed, the positive shock wave and the oblique shock wave affect the front edge of the wing and the front to the middle of the wing, respectively, and produce a complex waveform structure that oscillates up and down on the wing (Fig. R43)²². We found that the reason why DATSS can work normally in shock wave environment is that the position after the separation point of air flow is not greatly affected by shock wave, but the churning air still exists, so we made the above conjecture.

Fig. R43 Schematic diagram of airfoil flow separation in shock wave environment.

Are there any flaws in the data analysis, interpretation and conclusions? - Do these prohibit publication or require revision?

Effects of the device attached on the wing on aerodynamic performance might be required. Especially, in the case of realistic flight Reynolds number flow of the commercial aircrafts, the boundary layer becomes thin and is expected to be affected by the existence of the devices.

Response: Thank you very much for your comment. As another reviewer also mentioned, as the sensor is based on fluttering motion, it would naturally affect the efficiency of the aircraft by affect the drag and lift of the wings. When designing the DATSS system, we have fully considered the

impact on the flight efficiency of the aircraft. For the influence of the flight efficiency of the aircraft, two factors are mainly considered, lift and drag.

(1) For drag, the introduction of DATSS system mainly affects friction resistance and interference resistance, while the influence of differential pressure resistance and induced resistance can be negligible. **The DATSS system should minimize the effects of drag.** Firstly, the interference resistance is discussed. DATSS adopts a streamlined design in configuration. The ultra-thin hollow alloy steel sheet can conform to the airfoil of the aircraft and minimize the generation of interference resistance. When the T-signal (triboelectric signal) is working, we optimized the length of the steel sheet to make the vibration amplitude within 1 cm (different airfoils can be optimized differently), the height (1 cm) is less than the height of some screws on the airfoil of commercial aircraft. The counter electrode is 100 μm thick, which produces less interference resistance. Next, we discuss the friction resistance. When designing the DATSS system, in order to reduce the frictional resistance, we selected a stainless steel material with a smooth surface (low friction coefficient), and designed a hollow structure to reduce the surface area and mass of the DATSS system. The quality of the entire hollow alloy steel sheet is 0.9 g. In addition, the stiffness of the steel sheet is low, and its natural state is a streamlined structure that fits the airfoil. This makes it aerodynamically better for airflow over the surface. We have added these analyses in the Supplementary Information of revised manuscript.

(2) For lift and pressure drag, **DATSS system acts like active vortex generator, that can slightly delay airflow boundary separation and reduce pressure drag.** When designing the DATSS system, we referenced a large number of aircraft design literature, from which we conceived a reasonable system structure that can increase a part of the lift (which may have very little impact, we explain from the theoretical side). i) Blocking effect, when the airfoil flow separation stalls, the rolled steel sheet can use its rolled up tail to block the reversely separated airflow while generating the P-signal (piezoelectric signal), so that this part of the airflow can be redirected back to the incoming flow direction of the aircraft. This blocking effect can play a part in delaying the stall. ii) Recovery airflow effect. In the early stage of airflow separation, the steel sheet that flutters up and down will exert downward force on the airflow below during the descending process, so that this part of the airflow delays the airflow separation from the airfoil (similar to the action of a vortex

generator)^{23,24}. We have added the above experimental conclusions to the Supplementary Information of revised manuscript, highlighted in yellow.

(3) **Experimental supplement.** To verify the above theory, we supplement the relevant experiments of the lift test. The lift force of the model was tested using a six-axis force balance in a recirculation wind tunnel. Six-axis force balance is a standard equipment used to test lift force of aircraft. The device is shown in **Fig. R44**, considering that gravity might cause some interference with the lift test, we installed the model wing perpendicular to the direction in which it operates, so that gravity and lift are perpendicular to each other without interference. We compared and tested the airfoil lift with four DATSS units and without DATSS, as shown in **Fig. R45**. It can be seen from the figure that even if the DATSS system is installed on the whole airfoil in array, it has almost no impact on the airfoil lift, and even more than the original lift at some data points (in the figure, the wind speed in the wind tunnel increased gradually over time, finally, the final lift value was compared).

Fig. R44 Photograph of the wind tunnel test of the six-axis force balance.

Fig. R45 The airfoil lift with four DATSS units and without DATSS.

(4) **CFD simulation.** In addition to the analysis and discussion from the experimental point of view, we also considered the influence of the existence of the sheet on the lift and drag of the aircraft wing from the point of view of CFD simulation. Here, we consider the most extreme case, that is, to study how much lift and drag changes the steel sheet can bring to the airfoil without blocking backflow and downpressure airflow (both of which are helpful for lift). At the same time, in order to demonstrate the arrayable installation of the DATSS system, we placed four steel sheets on the wing section for simulation.

i) CFD parameter setting details: the incoming flow velocity is 200 m/s, the chord length is 0.3 m, and the span length is 0.5 m. Reynolds number: 4,107,522 (which can be identified as a high Reynolds number problem). Wing: NACA0012. Here we analyze the influence of the installation of the steel sheet on the lift drag of the airfoil. Since NACA0012 is a symmetrical wing, the lift of the symmetrical airfoil is basically close to 0 at an angle of attack (AoA) of 0°, which has no practical research significance. Therefore, we choose three cases where the AoA is 6°, 8°, 10° for CFD simulation. The software used is ANSYS2020-R2.

ii) Description of simulation mesh convergence: when there is no steel sheet on the airfoil, five convergence curves are calculated as residual, lift coefficient, drag coefficient, lift and drag curves as shown in Fig. R46, R47, R48, R49 and R50. The airfoil model without steel sheet has a total mesh number of 2 million.

Fig. R46 CFD simulation convergence calculation.

Fig. R47 Lift coefficient curve without steel sheet.

Fig. R48 Lift curve without steel sheet.

Fig. R49 Drag coefficient curve without steel sheet.

Fig. R50 Drag curve without steel sheet.

iii) The convergence curves when steel sheets are installed on the airfoil are shown in **Fig. R51**, **R52**, **R53**, **R54** and **R55** (because the number of meshes of the 3D model in this simulation is huge and the trial calculation time is long, here we only show the first 2800 steps of the curve of the trial calculation, and the trend can be seen from the curve, and the subsequent specific calculations have been completed on the supercomputer). The total number of grids of the airfoil model with steel sheets is 10.55 million, and the height of the first layer of grids is 0.002 mm.

Fig. R51 The convergence curve with steel sheet.

Fig. R52 Lift coefficient curve with steel sheet.

Fig. R53 Lift curve with steel sheet.

Fig. R54 Drag coefficient curve with steel sheet.

Fig. R55 Drag curve with steel sheet.

iv) ICEM grid and fluent settings: the unstructured grid is used for division, and the size of the external flow field is 10 times the chord length. When meshing, the leading and trailing edges of the wing and the tip of the wing are locally refined. At the same time, the steel sheet needs to be encrypted for the model. Since the thickness of the steel sheet is ultra-thin and the distance to the upper airfoil is relatively close, the grid needs to be dense enough to ensure the restoration accuracy of the model. Boundary layers can also create problems that are difficult to generate because they are too close together. Therefore, we use the operation of making a boundary layer first, and then segmenting it to realize the generation of the boundary layer mesh. The grid diagram is shown in **Fig. R56, R57, R58, R59 and R60**. We can observe the position of the encrypted part and the boundary layer. According to the analysis of the calculation conditions, it is a high Reynolds number problem. It is roughly estimated that the grid height of the first layer is 0.002 mm, and there are 15 layers in total.

Fig. R56 Schematic diagram of CFD model grid distribution.

Fig. R57 Schematic diagram of CFD wing front grid distribution.

Fig. R58 Schematic diagram of DATSS systems grid distribution.

Fig. R59 Side view of CFD modeling grid distribution.

Fig. R60 A local diagram of the DATSS system modeling subdivision grid.

v) Software parameter setting (ANSYS2020-R2):

No.	Set	Sort	Parameter
1	Solver	Type	Pressure-Based
		Velocity formulation	Absolute
		Time	Steady
2	Models	Energy	On
		Viscous	SST-K-ω
3	Materials	Air	Ideal-gas
		Viscosity	Sutherland
4	Operating Conditions	Operation Pressure	101,325 Pa
5	Boundary Conditions	Mach Number	0.5877270830
		X-Component of Flow Direction	6°, 8°, 10°
		Y-Component of Flow Direction	6°, 8°, 10°
		Turbulence Intensity	5%
		Turbulent Viscosity Ratio	10

6	Solution Methods	Scheme	SIMPLEC
		Skewness Correction	0
		Gradient	Least Squares Cell Based
		Pressure	Second Order
		Density	Second Order Upwind
		Momentum	Second Order Upwind
		Turbulent Kinetic Energy	First Order Upwind
		Specific Dissipation Rate	First Order Upwind

vi) Calculation results:

AoA	No steel sheet installed				Steel sheet installed				lift coefficient (C _l) ratio	drag coefficient (C _d) ratio
	C _{l-1}	L ₁	C _{d-1}	D ₁	C _{l-2}	L ₂	C _{d-2}	D ₂	C _{l-1} /C _{l-2}	C _{d-1} /C _{d-2}
6°	0.401656	1517.2147	0.031661	119.59552	0.401226	1515.5920	0.031389	118.5695	1.001071715	1.008665456
8°	0.533488	2015.19929	0.0467828	176.71777	0.533979	2017.0561	0.0465215	175.7056	0.999080488	1.005616758
10°	0.650746	2458.13043	0.0669403	252.860878	0.641714	2424.0139	0.0659274	249.0345	1.014074806	1.015363870

The unit of force is N. The convergence trend of small AoA is stable, and the monitoring data is basically stable. The maximum residual value of the final calculation is on the order of 1E-3, and the monitoring data changes very slowly, which can be judged as convergent. Through the simulation analysis, it can be seen that the small enough sheet structure has little effect on the lift before the wing stalls, about 0.1%. If the width of the sheet is further reduced, the effect similar to that of a vortex generator can be achieved. By creating a vortex above the airfoil to accelerate the airflow velocity on the upper airfoil, the effect of increasing lift and reducing drag can be achieved. At the same time, we have analyzed in the preceding paragraphs that the downward pressure air effect of the steel sheet and the blocking backflow effect can have a positive impact on the lift force.

Furthermore, the DATSS system is much lighter and thinner compared with other mounted sensors that are an order of magnitude thicker and much heavier, in a large number of literatures²⁵⁻²⁷. We hope that our systematic analysis and experiments above can provide you with a detailed description of the influence of the DATSS system on the lift and drag of the aircraft.

Is the methodology sound? Does the work meet the expected standards in your field?

The methodologies of experiments look okay. But I don't think that 2D RANS are suitable for these cases considered in this paper and the accuracy of sensing and measuring limits should be presented.

Response: Thank you very much for your comment, the accuracy and limits of the DATSS system have been described in detail in the above-mentioned discussion. For 2D RANS in CFD simulation, we modified it and used 3D model for CFD simulation. As we know, CFD simulation of 3D model based on fluid-structure interaction requires a lot of computing time and a lot of grid processing time, but its discussion on the working principle of DATSS system is more meaningful than that of 2D model, and it can consider more the relationship among flow field, airfoil and DATSS system. Therefore, at your suggestion, we choose to try 3D model CFD simulation. After about three months of exploration and calculation, we report the following conclusions to you, and attach the detailed calculation process and video files.

Supplementary video 6 completely shows the whole process of 3D CFD simulation of DATSS system. A continuous T-signal is generated at AoA of 0° , a continuous P-signal is generated at 18° (**Fig. R61**), and the T-signal is lost. This result is consistent with the conclusion of wind tunnel experiment and can be used as an important basis for theoretical explanation of the working principle of DATSS system. We have added this part of the discussion to the revised manuscript and Supplementary Information, highlighted in yellow. In the flow field, due to the combination of Coanda effect, airflow entrainment and self-vibration of airfoil when the aircraft is not stalled, the airflow over the airfoil causes a constant T-signal to be generated stably. During this process, the Coanda effect is a very important flow phenomenon and is the internal cause of the lift problem, using the Coanda effect, the air flow can be consciously induced to generate an air flow velocity greater than the relative air velocity on the upper surface of the triangularly-arranged rectangular hollow alloy steel sheet, just as the wing obtains lift, the steel sheet obtains an upward lift due to the pressure difference between the upper and lower surfaces. At this time, the rising steel sheet sucks the surrounding air into the lower surface due to its light and thin stiffness, in this moment, the air flow rate sucked by the entrainment effect is faster than the upper surface, lead to the steel sheet obtain a downward pressure and moves downward. Then the entrainment effect is reduced, and the Coanda effect makes the steel sheet rise again, under such cycles, the T-signal exists stably. In addition, the self-vibration of airfoil as it moves in the flow field exacerbates the intensity of the T-

signal. After the stall occurs, the reverse force caused by air separation causes the steel sheet to roll over, resulting in the P-signal, and the T-signal is lost due to the loss of contact between the electrodes. At this point, the DATSS system can monitor the stall occurrence and depth *in situ*. **Supplementary Video 7 shows the whole working process of DATSS system when $AoA=0^\circ$ and 18° .**

Fig. R61 CFD simulation of fluid-structure interaction of DATSS system when the AoA of the aircraft is 18° .

Is there enough detail provided in the methods for the work to be reproduced?

Turbulent property at the inlet flow of CFD is missing.

Response: Thank you very much for your comment. According to your suggestion, we modified the CFD simulation method and adopted the 3D fluid-structure interaction method to carry out the new CFD simulation. The results have been shown in the previous discussion, and now we describe the simulation setup in detail so that the process can be better reproduced. We have provided details of all the parameters you mentioned in the Supporting Information of the revised manuscript.

(1) **The creation of 3D models.** The 3D model of the steel sheet of DATSS system and aircraft wing is shown in **Fig. R62**. The wing shape curve was the section curve of NACA0012. The parameters of the curve were obtained from the official website of airfoiltools (<http://airfoiltools.com/>). Based on the calculation amount, the wing model was finally determined to be 195 mm in chord length and 200 mm in span length. The size of the sheet body was 110 mm long, 20 mm wide and 0.4 mm thick. Its material was set as structural steel. The mounting position was 25 mm from the leading edge vertex of the wing and was centered above the wing. The specific parameters are shown as followed: the material density was 7850 kg/m^3 , Young's modulus was $2\text{E}+11 \text{ Pa}$, Poisson's ratio was 0.3.

Fig. R62 The 3D model of the steel sheet of DATSS system and aircraft wing.

(2) **CFD simulation conditions.** In order to simulate the real environment of the wing, the simulated incoming flow velocity (v) was set as 200 m/s and the Reynolds number (Re) was 2,669,889 at standard sea level temperature, y^+ was 30. The simulation calculated the motion state

of the DATSS at AoA of 0° and 18° respectively. The simulation used two-way fluid-structure interaction simulation, and the transient structural module of Ansys Fluent was used for calculation.

(3) **Pre-processing of models.** The fluid domain was generated using Fluent's Geometry module, whose size was 1500 mm long, 300 mm wide and 400 mm high. The model was placed at 1/3 of the length direction, and placed symmetrically in the Y-axis and Z-axis directions. For the convenience of grid division in the later period, the fluid domain was divided into blocks based on the mesh model of the wing and steel sheet (**Fig. R63**).

Fig. R63 Partition of fluid domain blocks.

The Mesh module of Fluent was used to generate the fluid domain grid. Entering the Mesh module, the solid part was suppressed first, and the fluid domain part was generated by grid. In fluid grid generation, face sizing command was used to control the mesh size of steel sheet, and the element size of steel sheet was 2.0 mm. The mesh size of fluid domain was generated using body sizing command, and the element size of fluid was 10.0 mm. The resulting grids are shown in **Fig. R64** and **R65**.

Fig. R64 Fluid domain grid generation.

Fig. R65 Enlarged view of 3D model mesh detail.

Create named selection was used to name the cross sections of the fluid domain so that the boundary conditions could be applied later. The corresponding surface of the leading edge of the wing was the fluid inlet surface (inset), and the corresponding surface of the trailing edge of the wing was the fluid outlet surface (outset). The upper, lower and left and right sides of the wing were

set as planes of symmetry (sym1-4). Then, the wing surface (yi) and the fluid-structure interaction surface (fsiwall) of the steel sheet were set respectively.

(4) **Fluent setting.** After the grid model was imported, the general item was set. The type item of solver module was set as pressure-based, the velocity formulation item was set as absolute, and the time item was set as transient. **Models setting:** opened the energy equation under the models module, selected SST *k*-omega model for turbulence model, and adopted Fluent default settings for other settings. **Materials setting:** in this module, the material in the fluid domain was set to air, the density was set to ideal-gas, and the other gas parameters were set by Fluent default settings. **Boundary conditions setting:** set corresponding boundary conditions in this module, and referred to the naming rules mentioned above for specific boundary conditions. Velocity-inlet was set at the boundary of fluid inlet, where the incoming flow speed was 200 m/s, and other inlet parameters were set by default. The outlet boundary was set to the pressure outlet. The plane of symmetry (sym1-4) was set as the symmetric boundary condition, and the wing boundary (yi) and the fluid-structure interaction surface (fsiwall) were both wall boundary conditions. **Dynamic mesh setting:** in this module, set specific parameters of dynamic mesh. In mesh methods item, selected smoothing and remeshing modes, and in options item, selected implicit update, contact and detection. Then selected the linearly elastic solid mode from the smoothing option in the mesh method settings module, selected the local cell mode in the remeshing option, and set the size remeshing interval in the parameters to 1, that was, grid update was calculated once per iteration. Set the contact gap between yi boundary and fsiwall boundary in the contact detection of the item of options, that was, the proximity threshold was 0.1 mm. Finally, the boundary of fsiwall was created as a fluid-structure interaction system coupling. **Solution methods setting:** set the relevant parameters of the solver in this module, **Fig. R66** shows the detailed settings. **Solution initialization setting:** inlet boundary conditions were used to initialize the fluid domain calculation. The related settings and results are shown in **Fig. R67**. **Solution run calculation setting:** set the time step size to 0.0001 and the number of time steps to 1. Since this simulation was a two-way fluid-structure interaction calculation, the calculation in the Fluent solver was not the actual calculation step size and calculation step. For the setting of relevant parameters, refer to the parameter setting in the following section.

Fig. R66 Solution methods setting.

Fig. R67 Solution initialization setting.

(5) **Transient structural setting. Pre-processing of the model:** set the relevant parameters of the fixed part, in the pre-processing part, the model related to the fluid domain needed to be suppressed first. **Connections setting:** in this part, the minimum distance between the airfoil on the wing and the steel sheet was set. The type was set to frictionless mode, and the minimum distance item “offset” was set to 0.1 mm. **Model grid generation:** body sizing was used to generate the mesh of the wing and the steel sheet respectively. This simulation focused on the influence of the DATSS in the fluid domain, so a small-size mesh was generated for the steel sheet mesh, and its element size was set to 2.0 mm. For the wing mesh, the element size of the wing was set to 20.0 mm, the final grid model is shown in **Fig. R68**. **Transient fixed support setting:** set the fixed constraint surface of the steel sheet, as shown in **Fig. R69**. **Transient fluid solid interface setting:** six surfaces of the steel sheet were set as fluid-structure interaction surfaces, and the results were shown in **Fig. R70**. **Transient fixed support setting:** set the wing surface as a fixed surface, and the result is shown in **Fig. R71**.

Fig. R68 3D model grid generation diagram.

Fig. R69 The fixed constraint surface of the steel sheet.

Fig. R70 The setting of fluid-structure interaction surfaces.

Fig. R71 Wing surface fixed constraint setting.

(6) **System coupling setting. Analysis settings:** set the simulation calculation termination time and calculation step length. The total calculation time of AoA=18° was 1 s, and the time step was 5E-05 (the total calculation time of AoA=0° was 2 s, and the time step was the same as AoA=18°). **Data transfers setting:** set the data transfer between the steel sheet and the fluid domain, the results are shown in **Fig. R72**.

Fig. R72 Data transfers setting.

In summary, we explored and answered your comments on Reynolds number from the perspective of theory, simulation and experiment. In addition, for your suggestion that 2D CFD is not suitable for explaining DATSS system, we did 3D CFD simulation of fluid-structure interaction (FSI) and highlighted it in the revised manuscript, which is very difficult and time-consuming simulation means. What is more, we have conducted a preliminary exploration of the feasibility of the DATSS system on real aircraft, so as to make the DATSS system more suitable for industrial development. We compared a large number of existing technologies and literature, and summarized the originality and advantages of the DATSS system. We hope that after we have modified and perfected the original work, the revised work can meet your expectations for our work. At the same

time, the DATSS experiment of manned civil aviation has been planned, and we hope to apply it to the industrial field in the shortest time.

Thank you again for your valuable comments and precious time. It has been a privilege to communicate with you and learn through *Nat. Commun.* and is also our honor to greatly improve our work. Very likely, on our view of thinking, we can carry out more in-depth cooperation and exchanges in many aspects such as fluid mechanics sensing.

References

- 1 Tauer TM, Kunz DL, Lindsley NJ. Visualization of nonlinear aerodynamic phenomena During F-16 limit-cycle oscillations. *Journal of Aircraft* **53**, 865-870 (2016).
- 2 Strader J, Harper S, Gu Y. Aircraft Instrumentation and Computer Vision-Aided Flight Analysis of Local Air Flow. *AIAA Flight Testing Conference* (2016).
- 3 DEL FRATE J, SALTZMAN J. In-flight flow visualization results from the X-29A aircraft at highangles of attack. *6th AIAA Biennial Flight Test Conference* (1992).
- 4 Wadcock AJ, Ewing LA, Solis E, Potsdam M, Rajagopalan G. Rotorcraft downwash flow field study to understand the aerodynamics of helicopter brownout. (2008).
- 5 Lamar JE. Some vortical-flow flight experiments on slender aircraft that impacted the advancement of aeronautics. *Progress in Aerospace Sciences* **45**, 147-168 (2009).
- 6 Mack M, McMasters J. High Reynolds number testing in support of transport airplane development. *17th Aerospace ground testing conference* (1992).
- 7 Timmer W. Two-dimensional low-Reynolds number wind tunnel results for airfoil NACA 0018. *Wind engineering* **32**, 525-537 (2008).
- 8 Du L, Berson A, Dominy RG. Aerofoil behaviour at high angles of attack and at Reynolds numbers appropriate for small wind turbines. *Proceedings of the Institution of Mechanical Engineers, Part C: Journal of Mechanical Engineering Science* **229**, 2007-2022 (2015).
- 9 Worasinchai S, Ingram G, Dominy R. A low-Reynolds-number, high-angle-of-attack investigation of wind turbine aerofoils. *Proceedings of the Institution of Mechanical Engineers, Part A: Journal*

- of *Power and Energy* **225**, 748-763 (2011).
- 10 Saha N. Gap size effect on low Reynolds number wind tunnel experiments. *Virginia Tech* (1999).
- 11 Kernstine K, Moore C, Cutler A, Mittal R. Initial Characterization of Self-Activated Movable Flaps," Pop-Up Feathers". *46th AIAA Aerospace Sciences Meeting and Exhibit* (2008).
- 12 Ito MR, Duan C, Chamorro LP, Wissa AA. A leading-edge alula-inspired device (lead) for stall mitigation and lift enhancement for low Reynolds number finite wings. *Smart Materials, Adaptive Structures and Intelligent Systems* (2018).
- 13 Castaneda D, Whiting N, Webb NJ, Samimy M. Design and characterization of an experimental setup for active control of dynamic stall over a NACA 0012 airfoil. *AIAA Aviation 2019 Forum* (2019).
- 14 Singhal A, Castañeda D, Webb N, Samimy M. Control of dynamic stall over a NACA 0015 airfoil using plasma actuators. *AIAA Journal* **56**, 78-89 (2018).
- 15 Selig MS, Guglielmo JJ. High-lift low Reynolds number airfoil design. *Journal of aircraft* **34**, 72-79 (1997).
- 16 Crowder J. Flow visualization in flight testing. *Orbital Debris Conference: Technical Issues and Future Directions* (1990).
- 17 Yarusevych S, Sullivan PE, Kawall JG. Smoke-wire flow visualization in separated flows at relatively high velocities. *AIAA journal* **47**, 1592-1595 (2009).
- 18 Sankaralingam L, Ramprasad C. A comprehensive survey on the methods of angle of attack measurement and estimation in UAVs. *Chinese Journal of Aeronautics* **33**, 749-770 (2020).
- 19 Xiong W, *et al.* Bio-inspired, intelligent flexible sensing skin for multifunctional flying perception. *Nano Energy* **90**, 106550 (2021).
- 20 Liu G, Jiang Y, Wu P, Ma Z, Chen H, Zhang D. Artificial Whisker Sensor with Undulated Morphology and Self-Spread Piezoresistors for Diverse Flow Analyses. *Soft Robotics* (2022).
- 21 Lv S, *et al.* Gas-enhanced triboelectric nanogenerator based on fully-enclosed structure for energy harvesting and sensing. *Nano Energy* **55**, 463-469 (2019).
- 22 Zhonghua H, Zhenghong G, Wenping S, Lu X. On airfoil research and development: history, current status, and future directions. *Acta Aerodynamica Sinica* **39**, 1-36 (2021).
- 23 Bechert D, *et al.* Biological surfaces and their technological application-laboratory and flight experiments on drag reduction and separation control. *28th Fluid dynamics conference* (1997).

24 Kundu PK, Cohen IM, Dowling DR. *Fluid mechanics*. (2015).

25 Sadraey MH. *Aircraft design: A systems engineering approach*. (2012).

26 Kim Y, Jeon Y-H, Lee D-H. Multi-objective and multidisciplinary design optimization of supersonic fighter wing. *Journal of aircraft* **43**, 817-824 (2006).

27 Raymer D. *Aircraft design: a conceptual approach*. (2012).

REVIEWER COMMENTS

Reviewer #1 (Remarks to the Author):

The authors have answered all the comments in detail in the revised manuscript. I recommend this paper to be published as is.

Reviewer #2 (Remarks to the Author):

The authors have well addressed all my concerns. The revised manuscript is recommended to be accepted.

Reviewer #3 (Remarks to the Author):

1) Thanks for conducting the extra test. Airfoil geometry and angle of attack of the extra test were missing. It was difficult for me to imagine the flow field around the airfoil for the extra test. In addition, quantitative discussion about the influences of the Reynolds number on experimental results was required.

2) Since insufficient experimental testing for high Reynolds number flows, I recommend the authors to focus on aircrafts that have flown in intermediate Reynolds numbers and modify the introduction accordingly. Also, I recommend the authors to submit this manuscript to appropriate different journals.

3) Thanks for offering the literature review in the field of stall sensing. (3) and (4) were relative. I was expected constructive comparison of the system proposed in the manuscript and (4) in this revision.

4) Regarding the self-similarity region, is it applicable to the flow-fields around the airfoils? If so, please provide the evidence and/or references.

5) Fig.R43: Airfoil shape, angle of attack, Reynolds number, and Mach number were missing.

6) Providing impacts of the device on lift and drag were important for not only basic performance but also control and stability of the aircrafts.

7) Thanks for conducting very difficult and time-consuming 3D CFD simulation of FSI. But a lot of information were missing regarding the computation. I worried about the accuracy of the simulation presented in the manuscript.

REVIEWER COMMENTS

Reviewer #1 (Remarks to the Author):

The authors have answered all the comments in detail in the revised manuscript. I recommend this paper to be published as is.

Response to Reviewer 1: We greatly appreciate the time and valuable comment of the reviewer.

Reviewer #2 (Remarks to the Author):

The authors have well addressed all my concerns. The revised manuscript is recommended to be accepted.

Response to Reviewer 2: We greatly appreciate the time and valuable comment of the reviewer.

Reviewer #3 (Remarks to the Author):

Response to Reviewer 3: We greatly appreciate the time and valuable comments of the reviewer.

1) Thanks for conducting the extra test. Airfoil geometry and angle of attack of the extra test were missing. It was difficult for me to imagine the flow field around the airfoil for the extra test. In addition, quantitative discussion about the influences of the Reynolds number on experimental results was required.

Response: Thank you very much for your comment. The specific parameters of airfoil geometry are shown in **Fig. R1**. The airfoil is NACA0012, with a chord length of 600 mm and a width of 422.60 mm. The angle of attack (AoA) can be changed through a circular rotating shaft in the

middle of airfoil as shown in the figure. T/P-signal is the whole process signal when the AoA changes from 0 degrees to 16 degrees and then to 0 degrees. For the flow field around the airfoil, 2D CFD and 3D FSI simulations were carried out for the flow direction and size of the airfoil surface in the previous response (Fig. R2, Fig. R3), the exact setup down to each step is described in the answer to the seventh question. It is difficult to visualize flow fields with millions of Reynolds numbers experimentally.

Fig. R1 Specific parameters of airfoil geometry.

Fig. R2 CFD simulation of the flow state around the airfoil surface when the AoA of the aircraft increased from 0° to 24°.

Fig. R3 CFD simulation of fluid-structure interaction of DATSS system when the AoA of the aircraft is 18° .

For quantitative discussion about the influences of the Reynolds number, we conducted additional experiments to investigate. In this part, we do an in-depth analysis of the T/P-signal. For the T-signal under different AoA and different wind speeds, we have done a more systematic study, as shown in **Fig. R4**. After further study, we found that there is a relationship between the frequency of T-signal and **the Reynolds number**, in other words, in the DATSS system, we can also judge the current wind speed (flight speed) according to the T-signal, which makes the function of the DATSS system has been increased. At the same time, we also find that there are many reports on the correlation between the vibration frequency of T-signal and wind speed, but there are few reports under high wind speed. Therefore, this conclusion can supplement the signal characteristics of T-signal under high wind speed. But for the T-signal amplitude, the test uses a Keithley 6514 electrometer for a standard test, and its amplitude increases slightly with wind speed. In addition, we tested the frequency change of T-signal with AoA of 0° to 10° at the same

wind speed. It was found that at the Reynolds number of $\sim 4.5 \times 10^5$, the frequency of T-signal was all around ~ 40 Hz, and no correlation was found.

Fig. R4 The relationship between T-signal frequency/amplitude and the Reynolds number.

At the same time, we measured the P-signal peak value under different bending angles several times, as shown in **Fig. R5**. The stall depth can be roughly judged according to the amplitude of the P-signal. In future commercial products, the relative change of the P-signal amplitude may be used to analyze the degree of the stall, however, it is necessary to overcome the interference of signal in complex turbulent environment. We can also see the original data of P-signal in revised manuscript Fig. 3d, the amplitude of P-signal changed greatly in the turbulence of stall. We can only judge the degree of stall by comparing the P-signal value corresponding to its bending angle at a certain time under **different Reynolds numbers**. We have put these discussions into Supplementary Information of revising the manuscript. Besides, the fluctuation ranges of P-signal amplitudes measured at different Reynolds numbers are listed as follows.

Table 1. Fluctuation ranges of P-signal amplitudes at different Reynolds numbers.

Reynolds numbers	Fluctuation ranges of P-signal

558659.2	0-2 V
837988.8	0-4 V
3486033.5	0-10 V

Fig. R5 The P-signal peak value under different bending angles.

2) Since insufficient experimental testing for high Reynolds number flows, I recommend the authors to focus on aircrafts that have flown in intermediate Reynolds numbers and modify the introduction accordingly. Also, I recommend the authors to submit this manuscript to appropriate different journals.

Response: Thank you very much for your comment. In the revised manuscript, we marked the specific Reynolds number involved in each experiment without using words such as high and low Reynolds number. At the same time, according to your suggestion, we point out that our experimental range is in the range of intermediate Reynolds numbers in revised manuscript as: Here we report a non-invasive and lightweight active system that can sense and warn the pre-stall and during stall of fixed-wing aircraft in **intermediate Reynolds numbers** by employing conjunct signals as provided by triboelectric and piezoelectric effects.

Readers of *Nature Communications* may be readers with a wide range of knowledge background. Therefore, we have added the test data of **real Cessna C172S aircraft** in this reply. We hope that you can evaluate the suitability of this article for publication in *Nature Communications* again. Real aircraft flight testing requires contacting airlines, formulating flight plans, testing feasibility analysis of DATSS system, purchasing routes, installing DATSS system, installing data receivers, determining flight time and flight testing, which are complicated and difficult processes. We believe that the DATSS system can give you and your readers a deeper understanding after actual flight testing, **completely solve the problem of practical flight feasibility**.

Specific flight information is as follows (Table 2). DATSS system test data on a real Cessna C172S is shown in **Fig. R6**. The Reynolds number for cruising flight is about 5.4×10^6 , when AoA is 16° , the Reynolds number at flight is about 4.3×10^6 . The descending process of the T-Signal and the appearance process of the P-Signal correspond to the wind tunnel test data (**Fig. R6a**). **Fig. R6b** shows the photograph of real Cessna C172S aircraft and airport information. **Fig. R6c** exhibits the photograph of the DATSS system on the wing surface of a real Cessna C172S and inset is a takeoff test moment.

Table 2. Flight information.

Item	Information
Flight Date	February 2, 2023 6:00 pm
Location	Heze airport, Heze city
Longitude and Latitude	Longitude: 115.7° east; Latitude 35.2° north
Weather	Cloudy
Wind Direction	Northeast to South
Wind Speed	3-6 m/s
Visibility	4-10 km
Air Temperature	-3°C - 5°C
Aircraft	Cessna C172S
Pilot	Chenglei Wang

Flight Speed	80-100 knot (41.15-51.44 m/s)
Flight Altitude	Under 900 m
AoA of Stall Flight	16° (P-Signal)
AoA of Level Flight	0° (T-Signal)
Reynolds Number	$\sim 4.3 \times 10^6 - 5.4 \times 10^6$
Wingspan	36 feet 1 inch
Body Length	27 feet 2 inch
Tail Wingspan	11 feet 4 inch

Fig. R6 (a) The T/P-signal test of DATSS system in real Cessna C172S aircraft. (b) Photograph of real Cessna aircraft and airport information. (c) Photograph of the DATSS system on the wing surface of a real Cessna C172S and a takeoff test moment.

3) Thanks for offering the literature review in the field of stall sensing. (3) and (4) were relative. I was expected constructive comparison of the system proposed in the manuscript and (4) in this revision.

Response: Thank you very much for your comment. According to your suggestions, we have added a detailed constructive comparison description to the revised manuscript, compared with other stall solutions, and highlighted them.

According to the literature we have reviewed, here, we offer constructive comparison with other stall sensors, such as wind vane, air pressure, hot-film and piezoelectric sensors.

For wind vane AoA test method, compared to the DATSS system, the disadvantages are: i) indirect measurement, by turning the wind vane to a fixed critical AoA position, to warn the occurrence of stall. However, we know that the critical AoA is not a constant value, and it has a positive correlation with the flight speed and Reynolds number. The higher the flight speed, the higher the Reynolds number, result in the larger stall AoA. Therefore, it is obvious that there are some inaccuracies in the stall warning of the most commercially available wind vane method. ii) The size and weight of the equipment. Take, for example, the SMV-1 AoA wind vane sensor from Simtec Buerger AG (Swiss), which is dedicated to drones. It is a device with a smaller size and weight among the existing sensor solutions. Its length is 16 cm, width is 5.3 cm, and the weight of the sensor is about 30 g. Compared with the DATSS system's 10 cm length, less than 0.1 mm thickness, and weight of the sensor is about 1 g, the DATSS system has advantages in size and quality. iii) Way to install. The installation of the wind vane sensor needs to be intrusive on the surface of the aircraft body, which can only be installed during the production of the aircraft, and later maintenance and replacement needs to be carried out by professionals. The DATSS system, by contrast, is more flexible in its tape-on installation, which was also demonstrated in flight tests of the Cessna C172S. iv) The working environment. Wind vane AoA sensing requires force to deflect ~30 g wind vane so as to carry out sensing work. Therefore, when the UAV flies at a low speed, it cannot work due to force problems. By contrast, the 1g DATSS system works at a minimum speed of ~10m/s in wind tunnel tests. v) Principles of stall sensing. The wind vane AoA sensor is generally installed outside the cockpit of aircraft. It can indirectly measure the stall through AoA, which is a non-*in-situ* measurement principle and cannot directly sense the airfoil air separation. In contrast, the DATSS system is *in situ* sensing the degree of airflow separation on

the airfoil, which is a more accurate stalling sensing method in principle. In addition, T/P signals are self-powered sensing signals that do not require external power supply, which is impossible for wind vane AoA sensing.

In Xiong et al.,^[1] they used various sensing skin techniques to detect stall, including pressure sensor, piezoelectric vibrating sensor, and heat-loss hot film sensor. For the pressure sensor, the signal gradually increases with AoA until it reaches 14 degree then decrease drastically after 16 degree. The problem is that the sensor has low output after a certain degree and we can't determine whether it is in stall by just checking the current data. Instead, a long period time of data is needed to determine whether it is in full stall. The hot-film and piezoelectric sensors also have the same problem. Besides the stall sensing signal under ideal scenarios, they also have their own problems, which cause false alarm, according to their measuring mechanism, when large airborne particles hit the sensor, the pressure sensor will have similar response as stall; the hot-film is affected by temperature change; the piezoelectric fluttering sensor will false alarm stall for other aerodynamic scenarios that will cause the airfoil fluttering. Comparing with their sensors, our sensor is specifically designed to be only sensitive to the reversal flow, which is unique to stall. Our sensor responses to all states of stall, where above-mentioned sensors only response to several AoA and don't response at all to the full stall states.

As in Na et al.,^[2] total 10 sensors (5 pressure and 5 flow sensors) were used to monitor stall. In addition, a machine learning algorithm was needed to process the data, although there is no indication of stall from the original data of 10 sensors and how the data is processed by the algorithm. Comparing with their sensor, our in situ sensor needed only one unit (can be arrayed) and needed no complicated algorithm, giving our sensor much more advantages on mechanism and application.

References:

- [1] Na X, Gong Z, Dong Z, Shen D, Zhang D, Jiang Y. Flexible skin for flight parameter estimation based on pressure and velocity data fusion. *Advanced Intelligent Systems* **4**, 2100276 (2022).
- [2] Xiong W, et al. Bio-inspired, intelligent flexible sensing skin for multifunctional flying perception. *Nano Energy* **90**, 106550 (2021).

4) Regarding the self-similarity region, is it applicable to the flow-fields around the airfoils? If so, please provide the evidence and/or references.

Response: Thank you very much for your comment on the self-similarity. As our response in **Point 2)**, we have modified the focus of the manuscript to intermediate Reynolds numbers according to the your suggestions. Thus, the self-similarity discussion in the manuscript is not necessary anymore and excluded in the manuscript.

Still, we have gathered the references about self-similarity on related scenarios for your reference:

(1) In section 5 in Pope's book,^[1] they discuss flow self-similarity and various self-similar flows including boundary layer shear flow, which is related with airfoil condition.

(2) In chapter 18 in Anderson's book,^[2] they discuss the numerical self-similar solution for the compressible, laminar flow over a flat plate, which is also airfoil related condition.

(3) Maciel et al., 2006 summarizes self-similarity of velocity defect and Reynolds stress profile under different Reynolds numbers.^[3] The momentum-thickness Reynolds numbers are up to 97,200.

(4) In Hoseinzadeh et al., 2020,^[4] the self-similar velocity profile is found and discussed in details. The experiments are done for five different values of the attack angle as well three magnitude for the jet flow velocity, up to 32.2 m/s.

(5) In the chapter of Flow Similarity in Wang's book,^[5] he wrote that when the Reynolds number is large, the eddy viscosity is dominant, and the molecular viscosity is almost negligible. Therefore, when the Reynolds number is greater than a certain value, the flow is roughly similar, which can be considered to be between 10^5 and 10^7 . The illustrating picture below the text shows that the pressure distribution on the surface of the real airfoil and the model airfoil is similar to that of the prototype.^[5]

References:

[1] Anderson J. EBOOK: Fundamentals of Aerodynamics. *McGraw hill* (2011).

[2] Pope SB, Pope SB. Turbulent flows. *Cambridge university press* (2000).

[3] Maciel Y, Rossignol K-S, Lemay J. Self-similarity in the outer region of

adverse-pressure-gradient turbulent boundary layers. *AIAA journal* **44**, 2450-2464 (2006).

[4] Hoseinzadeh S, Bahrami A, Mirhosseini SM, Sohani A, Heyns S. A detailed experimental airfoil performance investigation using an equipped wind tunnel. *Flow Measurement and Instrumentation* **72**, 101717 (2020).

[5] Wang, H. A Guide to Fluid Mechanics. *Cambridge University Press* (2023).

5) Fig.R43: Airfoil shape, angle of attack, Reynolds number, and Mach number were missing.

Response: Thank you very much for your comment. This is a schematic diagram of the flow field around the airfoil under shock waves mentioned **in the reference** (Han Z. H., Gao Z. H., Song W. P., et al. On airfoil research and development: history, current status, and future directions[J]. *Acta Aerodynamica Sinica*, 2021, 39(6): 1 – 36. Fig. 57: Complex flow structures of shock waves-boundary layer interaction under high-subsonic and low Reynolds number). In this reference, and in the references mentioned by the authors, the authors do not provide parameters such as AoA Reynolds number and Mach number, only two conditions of high subsonic velocity and low Reynolds number can be obtained, and AoA is about 0 degrees.

6) Providing impacts of the device on lift and drag were important for not only basic performance but also control and stability of the aircrafts.

Response: Thank you very much for your comment. We fully agree the comment “Providing impacts of the device on lift and drag were important for not only basic performance but also control and stability of the aircrafts”, therefore, after additional experiments of lift comparison and CFD lift comparison, we also conducted handling and stability tests of DATSS system on UAV and DATSS system on real Cessna C172S aircraft.

For control and stability, the UAV controller did not give any warning during the DATSS system test, at the same time, the operator has no subjective difference in the operation of the UAV after carrying the DATSS system. In the real Cessna C172S test, pilot Chenglei Wang did not receive any abnormal warning and alarm of the aircraft while carrying DATSS system, and

subjectively there was no difference compared with that without carrying DATSS system. So this is in line with your comment.

7) Thanks for conducting very difficult and time-consuming 3D CFD simulation of FSI. But a lot of information were missing regarding the computation. I worried about the accuracy of the simulation presented in the manuscript.

Response: Thank you very much for your comment. Here we have described the simulation setup in detail so that the process can be better reproduced. **We have provided details of all the parameters you mentioned in the Supporting Information of the revised manuscript as follows, without skipping a single step.**

(1) **The creation of 3D models.** The 3D model of the steel sheet of DATSS system and aircraft wing is shown in **Fig. R7**. The wing shape curve was the section curve of NACA0012. The parameters of the curve were obtained from the official website of airfoiltools (<http://airfoiltools.com/>). Based on the calculation amount, the wing model was finally determined to be 195 mm in chord length and 200 mm in span length. The size of the sheet body was 110 mm long, 20 mm wide and 0.4 mm thick. Its material was set as structural steel. The mounting position was 25 mm from the leading edge vertex of the wing and was centered above the wing. The specific parameters are shown as followed: the material density was 7850 kg/m³, Young's modulus was 2E+11 Pa, Poisson's ratio was 0.3.

Fig. R7 The 3D model of the steel sheet of DATSS system and aircraft wing.

(2) **CFD simulation conditions.** In order to simulate the real environment of the wing, the simulated incoming flow velocity (v) was set as 200 m/s and the Reynolds number (Re) was 2,669,889 at standard sea level temperature, y^+ was 30. The simulation calculated the motion state of the DATSS at AoA of 0° and 18° respectively. The simulation used two-way fluid-structure interaction simulation, and the transient structural module of Ansys Fluent was used for calculation (**Fig. R8**).

Fig. R8 Ansys Workbench calculates flowchart.

(3) **Pre-processing of models.** The fluid domain was generated using Fluent's Geometry module, whose size was 1500 mm long, 300 mm wide and 400 mm high. The model was placed at 1/3 of the length direction, and placed symmetrically in the Y-axis and Z-axis directions. For the convenience of grid division in the later period, the fluid domain was divided into blocks based on the mesh model of the wing and steel sheet (**Fig. R9**).

Fig. R9 Partition of fluid domain blocks.

The Mush module of Fluent was used to generate the fluid domain grid. Entering the Mush module, the solid part was suppressed first, and the fluid domain part was generated by grid (**Fig. R10**).

Fig. R10 The solid part is suppressed.

In fluid grid generation, face sizing command was used to control the mesh size of steel sheet, and the element size of steel sheet was 2.0 mm. The mesh size of fluid domain was generated using body sizing command, and the element size of fluid was 10.0 mm. The resulting grids are shown in **Fig. R11** and **R12**.

Fig. R11 Fluid domain grid generation.

Fig. R12 Enlarged view of 3D model mesh detail.

Create named selection was used to name the cross sections of the fluid domain so that the boundary conditions could be applied later. The corresponding surface of the leading edge of the wing was the fluid inlet surface (inset), and the corresponding surface of the trailing edge of the wing was the fluid outlet surface (outset). The upper, lower and left and right sides of the wing were set as planes of symmetry (sym1-4). Then, the wing surface (yi) and the fluid-structure interaction surface (fsiwall) of the steel sheet were set respectively.

(4) **Fluent setting.** After the grid model was imported, the general item was set. The type item of solver module was set as pressure-based, the velocity formulation item was set as absolute, and the time item was set as transient (**Fig. R13**).

Fig. R13 General setting.

Models setting: opened the energy equation under the models module, selected SST k - ω model for turbulence model, and adopted Fluent default settings for other settings (Fig. R14).

Fig. R14 Models setting.

Materials setting: in this module, the material in the fluid domain was set to air, the density was set to ideal-gas, and the other gas parameters were set by Fluent default settings (Fig. R15).

Fig. R15 Materials setting.

Boundary conditions setting: set corresponding boundary conditions in this module, and referred to the naming rules mentioned above for specific boundary conditions. Velocity-inlet was set at the boundary of fluid inlet, where the incoming flow speed was 200 m/s, and other inlet parameters were set by default. The outlet boundary was set to the pressure outlet. The plane of

symmetry (sym1-4) was set as the symmetric boundary condition, and the wing boundary (yi) and the fluid-structure interaction surface (fsiwall) were both wall boundary conditions (Fig. R16 and R17).

Fig. R16 Velocity-inlet setting.

Fig. R17 Pressure-outlet setting.

Dynamic mesh setting: in this module, set specific parameters of dynamic mesh. In mesh methods item, selected smoothing and remeshing modes, and in options item, selected implicit update, contact and detection (Fig. R18).

Fig. R18 Dynamic mesh setting.

Then selected the linearly elastic solid mode from the smoothing option in the mesh method settings module, selected the local cell mode in the remeshing option, and set the size remeshing interval in the parameters to 1, that was, grid update was calculated once per iteration. Set the contact gap between yi boundary and fsiwall boundary in the contact detection of the item of options, that was, the proximity threshold was 0.1 mm. Finally, the boundary of fsiwall was created as a fluid-structure interaction system coupling (**Fig. R19-22**).

Fig. R19 Smoothing setting.

Fig. R20 Remeshing setting.

Fig. R21 Options setting.

Fig. R22 Fluid-structure interaction setting.

Solution methods setting: set the relevant parameters of the solver in this module, **Fig. R23** shows the detailed settings. **Solution initialization setting:** inlet boundary conditions were used to initialize the fluid domain calculation. The related settings and results are shown in **Fig. R24**. **Solution run calculation setting:** set the time step size to 0.0001 and the number of time steps to 1. Since this simulation was a two-way fluid-structure interaction calculation, the calculation in the Fluent solver was not the actual calculation step size and calculation step. For the setting of relevant parameters, refer to the parameter setting in the following section (**Fig. R25**).

Fig. R23 Solution methods setting.

Fig. R24 Solution initialization setting.

Fig. R25 Solution run calculation setting.

(5) **Transient structural setting. Pre-processing of the model:** set the relevant parameters of the fixed part, in the pre-processing part, the model related to the fluid domain needed to be suppressed first. **Materials setting:** set the material parameters of the piece body, as shown in **Fig. R26**.

Fig. R26 Materials setting.

Connections setting: in this part, the minimum distance between the airfoil on the wing and the steel sheet was set. The type was set to frictionless mode, and the minimum distance item “offset” was set to 0.1 mm (Fig. R27).

Fig. R27 Connections setting.

Model grid generation: body sizing was used to generate the mesh of the wing and the steel sheet respectively. This simulation focused on the influence of the DATSS in the fluid domain, so a small-size mesh was generated for the steel sheet mesh, and its element size was set to 2.0 mm. For the wing mesh, the element size of the wing was set to 20.0 mm, the final grid model is shown in **Fig. R28**. **Transient fixed support setting:** set the fixed constraint surface of the steel sheet, as shown in **Fig. R29**. **Transient fluid solid interface setting:** six surfaces of the steel sheet were set as fluid-structure interaction surfaces, and the results were shown in **Fig. R30**. **Transient fixed support setting:** set the wing surface as a fixed surface, and the result is shown in **Fig. R31**.

Fig. R28 3D model grid generation diagram.

Fig. R29 The fixed constraint surface of the steel sheet.

Fig. R30 The setting of fluid-structure interaction surfaces.

Fig. R31 Wing surface fixed constraint setting.

(6) **System coupling setting. Analysis settings:** set the simulation calculation termination time and calculation step length. The total calculation time of $\text{AoA}=18^\circ$ was 1 s, and the time step was $5\text{E}-05$ (the total calculation time of $\text{AoA}=0^\circ$ was 2 s, and the time step was the same as $\text{AoA}=18^\circ$).

Data transfers setting: set the data transfer between the steel sheet and the fluid domain, the results are shown in **Fig. R32**.

Fig. R32 Data transfers setting.

(7) **Convergence curve.** The convergence curves calculated for $AoA=0^\circ$ and 18° are shown in **Fig. R33** and **R34**.

Fig. R33 Calculated convergence curve for $AoA=0^\circ$.

Fig. R34 Calculated convergence curve for $AoA=18^\circ$.

(8) **Post-processing of the result file.** The Results module is used for post-processing. After importing the result file, the Plane command is used to draw the display plane (Plane 1), as shown in **Fig. R35**. The velocity cloud map distribution is displayed on the Plane 1 (**Fig. R36**).

Fig. R35 Drawing the display plane.

Fig. R36 The velocity cloud map distribution.

In the end, this work, in its current version, has not been possible without your help and guidance on details and research methods. We would like to express the sincerest thanks of all the authors to you.